# Solar reduction of carbon dioxide on copper-tin electrocatalysts with energy conversion efficiency near 20%

Jing Gao [1,4] ✉, Jun Li[1,4], Yuhang Liu[1,4], Meng Xia[1], Y. Zou Finfrock[2], Shaik Mohammed Zakeeruddin[1], Dan Ren [3,1] ✉ & Michael Grätzel [1] ✉

Copper catalysts modified with tin have been demonstrated to be selective for the electroreduction of carbon dioxide to carbon monoxide. However, such catalysts require the precise control of tin loading amount. Here, we develop a copper/tin-oxide catalyst with dominant tin oxide surface being formed via a spontaneous exchange reaction between sputtered tin and copper oxide. Even though the surface of this catalyst is tin-rich, it achieves an excellent performance towards carbon monoxide production in a flow cell. This contrasts with copper/tin-oxide prepared via atomic layer deposition since it yields selectivity towards carbon monoxide only on a copper-rich surface. Mechanism studies reveal that the tin sites on the tin-rich copper/tin-oxide surface achieve a suitable binding with adsorbed carbon monoxide under the presence of copper. Powered by a triple-junction solar cell, the copper/tin-oxide based electrolyzer sets a new benchmark solar-to-chemical energy conversion efficiency of 19.9 percent with a Faradaic efficiency of 98.9 percent towards carbon monoxide under simulated standard air mass 1.5 global illumination.

Solar-driven carbon dioxide ($CO_2$) reduction, known as "artificial photosynthesis"[1–3], has the potential of abating $CO_2$ emission thus mitigating the impact of global warming[4–6]. Electrochemical $CO_2$ reduction powered by photovoltaics (PV-EC) has so far been demonstrated for the most efficient production of carbonaceous molecules such as carbon monoxide (CO) and ethylene ($C_2H_4$)[7–9]. However, the most commonly used electrolyzer employs an H-cell configuration and the geometric current density is limited to a few tens of mA cm$^{-2}$ due to the poor solubility of $CO_2$ (ca. 34 mM at ambient pressure and room temperature) in aqueous electrolyte. The recently-developed flow electrolyzer enables a substantial improvement in $CO_2$ reduction reaction rate[10,11]. While PV-EC systems integrating a photovoltaic with a gas diffusion electrolyzer for efficient synthesis of chemical fuels are more appealing, they are yet to be developed[8]. Recently, Atwater and co-workers demonstrated a solar-to-CO conversion efficiency of 19.1%

with a Ag catalyst and a triple-junction photovoltaic cell[12]. Demonstration of such an efficient solar-driven system with an Earth-abundant catalyst for $CO_2$ reduction has not been achieved so far.

Cu modified with $SnO_x$ or metallic Sn ($Sn^0$) has recently emerged as a non-noble and selective electrocatalyst for $CO_2$ conversion to CO at low overpotentials[13–15]. The catalytic activity of the Cu-Sn based catalysts varies with their structure, surface speciation and chemical composition[16,17]. Trace amount of $SnO_x$ imbedded in a Cu matrix promotes the CO formation by weakening the *CO (* denotes the adsorbed species) adsorption on Cu sites[7,18]. However, upon slightly increasing the Sn content, the selectivity of Cu-Sn catalyst for $CO_2$ reduction sharply shifts from CO to formate ($HCOO^-$) due to the fact that the exposed $Sn^0$ sites favor a stronger $HCOO^-$ adsorption on the catalyst[18]. This selectivity shift is also observed on the CuO doped with atomic layer deposited (ALD) $SnO_2$ catalyst developed in this

[1]Laboratory of Photonics and Interfaces, École Polytechnique Fédérale de Lausanne, 1015 Lausanne, Switzerland. [2]Structural Biology Center, X-ray Science Division, Argonne National Laboratory, Lemont, IL 60439, USA. [3]Present address: School of Chemical Engineering and Technology, Xi'an Jiaotong University, Xi'an 710049, China. [4]These authors contributed equally: Jing Gao, Jun Li, Yuhang Liu. ✉e-mail: jing.gao@epfl.ch; dan.ren@xjtu.edu.cn; michael.graetzel@epfl.ch

work (CuO-SnO₂ _ALD_). In order to avoid this unwanted change in selectivity towards HCOO⁻ formation, the thickness of the SnO₂ overlayer has to be less than 2.7 nm.

Here, we develop a new Cu-SnO₂ catalyst where the SnO₂ species are formed via spontaneous exchange reaction (SER) between CuO substrate and sputtered Sn particles (Cu-SnO₂ _SER_). In spite of its Sn-rich nature as evidenced by X-ray photoelectron spectroscopy (XPS) and lead underpotential deposition (Pb UPD), Cu-SnO₂ _SER_ achieves nearly exclusive production of CO with the thickness of the Sn overlayer varying from 40 to 160 nm. Using X-ray absorption spectroscopy (XAS), electrochemical analysis and CO adsorption/stripping experiments, we reveal that the Cu-regulated SnO₂ _SER_ sites have a suitable binding for forming C-bound *CO species, rendering its selectivity and activity for CO generation. Finally, we directly wire a III-V InGaP₂/InGaAs/Ge triple-junction solar cell to a custom-built two-electrode

flow electrolyzer, setting a new record for solar-to-CO energy conversion efficiency.

## Results

### Synthesis and characterization of CuO, CuO-SnO₂ _SER_ and CuO-SnO₂ _ALD_

CuO nanowires (NWs) were prepared by electrochemical anodization of a Cu film supported on a gas diffusion electrode (GDE) in 3 M KOH, followed by annealing at 150 °C for 1 hr under ambient condition (Fig. 1a)[19]. Metallic Sn with expected thickness of 60 nm was then sputtered onto CuO NWs. The spontaneous oxygen transfer from CuO to metallic Sn leads to the formation of SnO₂ (Supplementary Note 1) and the sample is denoted as CuO-SnO₂ _SER_. On the other hand, ultrathin layers of SnO₂ were coated onto CuO via atomic layer deposition for the preparation of CuO-SnO₂ _ALD_. Three samples

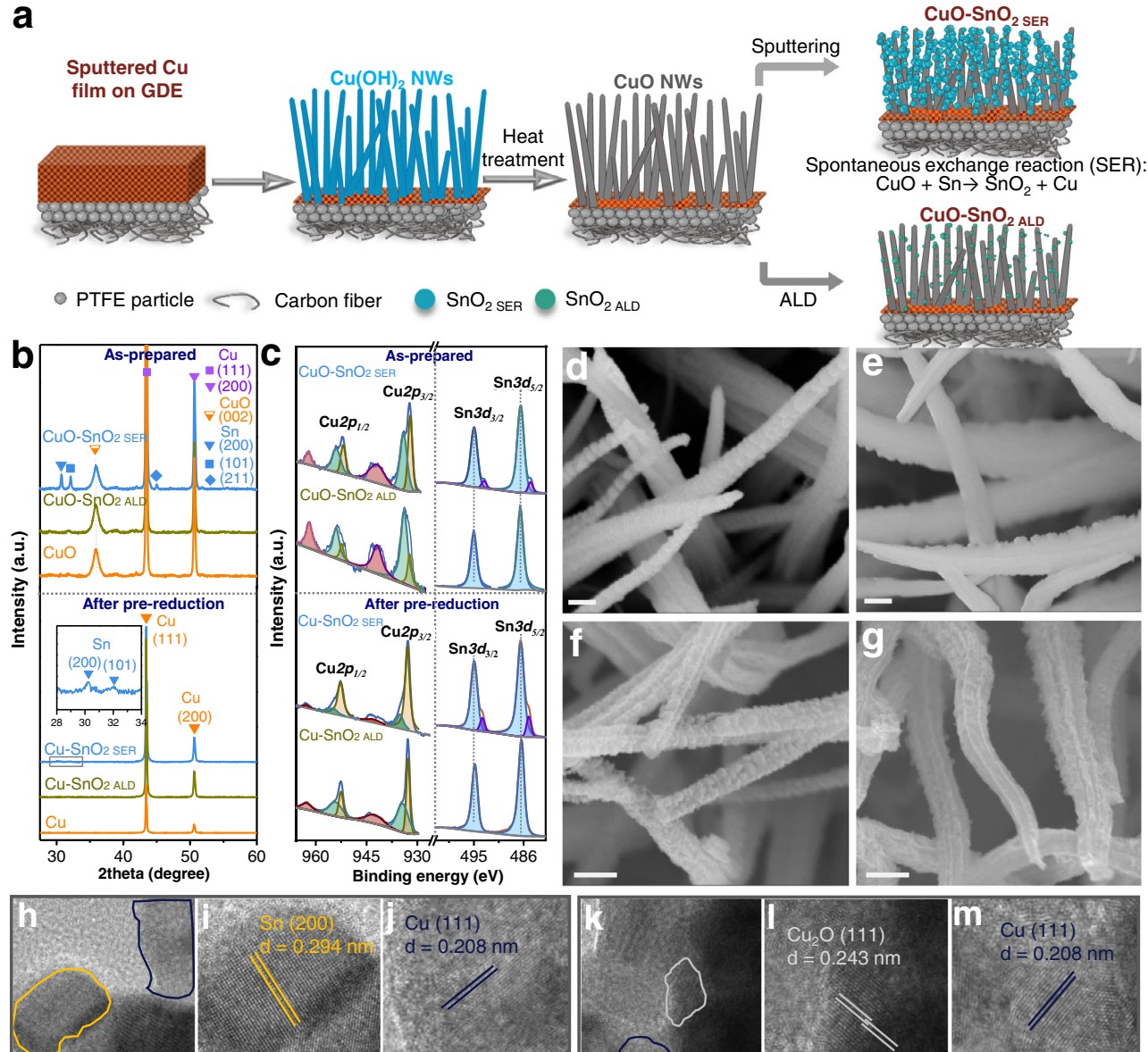

**Fig. 1 | Structural and chemical characterizations of Cu-SnO₂ catalysts. a** A schematic route for the synthesis of CuO-SnO₂ _SER_ and CuO-SnO₂ _ALD_ catalysts. **b** XRD and (**c**) XPS of as-prepared CuO, CuO-SnO₂ _SER_, CuO-SnO₂ _ALD_ and the ones after pre-reduction at −30 mA cm⁻² for -100 s. SEM images of as-prepared (**d**) CuO-SnO₂ _SER_, (**e**) CuO-SnO₂ _ALD_ samples and pre-reduced (**f**) Cu-SnO₂ _SER_, (**g**) Cu-SnO₂ _ALD_ catalysts. **h** Representative HRTEM images of pre-reduced Cu-

SnO₂ _SER_ and lattice fringes corresponding to (**i**) Sn (200) and (**j**) Cu (111) on Cu-SnO₂ _SER_ catalyst. **k** Representative HRTEM images of pre-reduced Cu-SnO₂ _ALD_ and lattice fringes corresponding to (**l**) Cu₂O (111) and (**m**) Cu (111) on Cu-SnO₂ _ALD_ catalyst. Scale bars: 200 nm for (d) to (g), 5 nm for (h) to (m). GDE, NWs and ALD denote gas diffusion electrode, nanowires and atomic layer deposition.

including CuO, CuO-SnO$_2$ $_{SER}$ and CuO-SnO$_2$ $_{ALD}$ were characterized by X-ray diffraction (XRD), XPS, scanning electron microscopy (SEM), and transmission electron microscopy (TEM) (Fig. 1b-i).

To avoid the interference from GDE, XRD of three samples that deposited on glass was performed (Fig. 1b). For all three samples, XRD reveals the existence of CuO. The detected peaks for metallic Cu are partially attributed to the underlying residual sputtered Cu[18]. CuO-SnO$_2$ $_{SER}$ exhibits visible peaks at 2θ = 30.8°, 32.2° and 44.9° that are assigned to diffractions from Sn(200), Sn(101) and Sn(211) facets, respectively (Fig. 1b)[20]. CuO-SnO$_2$ $_{ALD}$ shows no other peak than those from the diffractions of CuO and Cu even if the number of cycles for ALD was increased to 300 (Fig. 1b and Supplementary Fig. 1), indicating that the SnO$_2$ layer prepared by the ALD is amorphous[21]. The XRD patterns of all the samples prepared on the GDE are also provided in Supplementary Fig. 1.

As expected, CuO shows a Cu $2p$ peak at 934.1 eV in the XPS spectrum, representing its dominant Cu(II) species (Supplementary Fig. 2). CuO-SnO$_2$ $_{SER}$ shows Cu(I/0) $2p$ peaks at 932.6 and 952.3 eV as well as two intense peaks at 486.6 and 495.0 eV for Sn(IV) $3d_{5/2}$ and $3d_{3/2}$, respectively (Fig. 1c). This demonstrates that the spontaneous reaction between CuO and Sn leads to the formation of SnO$_2$ with trace amount of metallic Sn remaining. Note that the possible atmospheric oxidation of the sample surface was minimized by immediately vacuum packing after sputtering. CuO-SnO$_2$ $_{ALD}$ shows only Sn(IV) $3d_{5/2}$ and $3d_{3/2}$ at 486.7 and 495.1 eV, respectively, confirming the purity of Sn (IV) prepared by ALD (Fig. 1c and Supplementary Fig. 2)[22]. XPS survey spectra of three samples are shown in Supplementary Fig. 2.

SEM reveals a nanowire structure of CuO with diameters of 200−400 nm and lengths of several micrometers (Supplementary Fig. 3). Nanoparticles with sizes of 60-90 nm are observed on the surface of CuO-SnO$_2$ $_{SER}$ (Fig. 1d). The high-angle annular dark field (HAADF) image and energy dispersive X-ray (EDX) spectroscopic mapping of CuO-SnO$_2$ $_{SER}$ show a mixture of Cu/CuO and Sn/SnO$_2$ particles on the sample surface (Supplementary Fig. 4), visualizing the oxygen exchange between Sn and CuO. For CuO-SnO$_2$ $_{ALD}$, a 5-cycle ALD SnO$_2$ coating is invisible under SEM (Fig. 1e), while HAADF-EDX mapping, high resolution (HR)TEM images and SEM-EDX mapping demonstrate the distribution of SnO$_2$ on the surface of CuO NWs (Supplementary Figs. 5-7). The thickness of SnO$_2$ with 5-ALD-cycle is estimated to be 1.35 nm by a linear fitting between thickness and number of ALD cycles (Supplementary Figs. 8-9).

## Change of morphology and chemical composition after electrochemical reduction

The three samples were pre-reduced at a constant geometric current density of −30 mA cm$^{-2}$ for -100 s in a custom-designed flow cell with 0.5 M KHCO$_3$ and CO$_2$ being separately infused at flow rates of 0.25 and 20 cm$^3$ min$^{-1}$, respectively (Supplementary Fig. 10). They were then characterized by XRD, XPS, SEM, TEM and EDX.

For all three samples, disappearance of XRD peaks from CuO demonstrates the reduction of CuO substrate to Cu (Fig. 1b and Supplementary Fig. 11). Thus, the three samples after pre-reduction are denoted as Cu, Cu-SnO$_2$ $_{SER}$ and Cu-SnO$_2$ $_{ALD}$ afterwards. No alloy phase is observed by XRD on either Cu-SnO$_2$ $_{SER}$ or Cu-SnO$_2$ $_{ALD}$.

High-resolution XPS spectrum of Cu shows dominant metallic Cu features for Cu-SnO$_2$ $_{SER}$ and Cu-SnO$_2$ $_{ALD}$ (Fig. 1c). Sn(IV) $3d$ peaks for Cu-SnO$_2$ $_{SER}$ remain almost identical with the ones of CuO-SnO$_2$ $_{SER}$. However, the binding energy of Sn(IV) $3d$ peaks on Cu-SnO$_2$ $_{ALD}$ surface is 0.4 eV lower as compared to CuO-SnO$_2$ $_{ALD}$, which is attributed to the partial reduction of Sn$^{4+}$ to Sn$^{2+}$.[18] The persistence of SnO$_x$ species on both samples is consistent with previous reports[13,18,23]. This might be due to the partial electron transfer from Sn to Cu as proposed by Vasileff et al.[24]. According to our XPS analysis, Cu-SnO$_2$ $_{SER}$ shows a Sn-rich surface with a Sn-to-Cu ratio of 58:42, while the ratio on Cu-SnO$_2$ $_{ALD}$ surface is around 10:90 (Supplementary Table 1). This is consistent

with Pb UPD measurement, showing a 7-fold decrease in the number of Cu active sites on Cu-SnO$_2$ $_{SER}$ as compared to Cu (Supplementary Fig. 12 and Table 2)[25].

Cu NWs reconstruct into flakes after pre-reduction (Supplementary Fig. 11), resulting in a highly rough surface (Supplementary Fig. 13 and Supplementary Table 3). For Cu-SnO$_2$ $_{SER}$ catalyst, Sn/SnO$_2$ nanoparticles remain on the surface of Cu NWs (Fig. 1f). For Cu-SnO$_2$ $_{ALD}$, SnO$_2$ nanoparticles with size of around 10 nm are dispersed on the surface (Fig. 1g). SEM-EDX analysis reveals that the bulk concentration of Sn remains unchanged on both catalysts after pre-reduction (Supplementary Fig. 14).

HRTEM images and the selected area electron diffractograms show lattice fringe of 0.294 nm that is assigned to metallic Sn(200) on Cu-SnO$_2$ $_{SER}$ (Fig. 1h and i and Supplementary Fig. 15). The lattice fringe of metallic Cu(111) with a spacing of 0.208 nm is also visible on both Cu-SnO$_2$ (Fig. 1j and m). The observation of Cu$_2$O(111) facet is likely due to the oxidation of Cu during the preparation of TEM sample (Fig. 1k and l). HAADF-EDX analysis shows an even distribution of Sn on the surface of Cu NWs on both pre-reduced catalysts (Supplementary Fig. 16).

Additional characterizations of the catalysts after 50 min electrolysis at −50 mA cm$^{-2}$ are shown in Supplementary Figs. 17-18. Both Cu-SnO$_2$ catalysts show a noticeable decrease in Sn content after 50 min-reaction, mainly due to the partial dissolution of Sn into electrolyte. This is further corroborated by the ICP-OES analysis of the electrolyte that was collected after electrolysis on Cu-SnO$_2$ $_{SER}$ catalyst (Supplementary Table 4). However, the amount of Sn dissolved from Cu-SnO$_2$ $_{ALD}$ is not detectable by ICP-OES since the loading of SnO$_2$ $_{ALD}$ is extremely low.

## Selective and durable electrosynthesis of CO on Cu-SnO$_2$ $_{SER}$ and Cu-SnO$_2$ $_{ALD}$

The catalytic performances of Cu-SnO$_2$ $_{SER}$ and Cu-SnO$_2$ $_{ALD}$ as well as Cu catalysts were analyzed in 0.5 M KHCO$_3$ using a custom-designed flow reactor. Linear sweep voltammograms reveal that Cu-SnO$_2$ $_{SER}$ and Cu-SnO$_2$ $_{ALD}$ exhibit lower cathodic geometric current density than Cu (Supplementary Fig. 19). For example, the cathodic current density on Cu reaches -195 mA cm$^{-2}$ at -0.80 V vs. reversible hydrogen electrode (RHE), which is ca. 2.2× and 1.5× larger than the ones on Cu-SnO$_2$ $_{SER}$ and Cu-SnO$_2$ $_{ALD}$, respectively.

We performed chronopotentiometric measurements at different current densities from −20 to −250 mA cm$^{-2}$ with gaseous products being analyzed by an online gas chromatography and liquid products being evaluated by a high-performance liquid chromatography (Supplementary Note 2, Supplementary Tables 5-12). The total Faradaic efficiency (FE) of products on Cu is well below 100% at current densities from −20 to −100 mA cm$^{-2}$ (Supplementary Table 5), which is due to the crossover of liquid products in ionic form from the catholyte to the anolyte (Supplementary Table 6)[26]. Cu catalyzes the production of a mixture of carbonaceous products, with H$_2$ being dominant if the applied cathodic current density is less than 60 mA cm$^{-2}$ (Supplementary Fig. 20). Once the cathodic current density reaches 250 mA cm$^{-2}$, C$_{2+}$ products are favored with a peak FE of 75%.

Cu-SnO$_2$ $_{ALD}$ delivers a remarkable FE of 99% for CO generation at −40 mA cm$^{-2}$ (Fig. 2a). On Cu-SnO$_2$ $_{SER}$, FE for CO production increases from 64% at −20 mA cm$^{-2}$ to 98% at −50 mA cm$^{-2}$ and remains >83% till the current density reaches −100 mA cm$^{-2}$ (Fig. 2b). Notably, the Faradaic yield of H$_2$ is suppressed by up to an order of magnitude on both catalysts as compared to Cu (Supplementary Tables 9 and 11). As the cathodic current density further increases to 150−250 mA cm$^{-2}$, C$_{2+}$ products including ethylene, ethanol and $n$-propanol are formed on both Cu-SnO$_2$ catalysts. A FE$_{C_{2+}}$ of 12.5−14.6% is observed on Cu-SnO$_2$ $_{SER}$, while ethanol and trace amount of ethylene are produced by Cu-SnO$_2$ $_{ALD}$, with a maximum FE$_{C_{2+}}$ of only 2.3% (Supplementary Fig. 20). The FEs of different products on Cu-SnO$_2$ $_{ALD}$ and Cu-SnO$_2$ $_{SER}$ are also

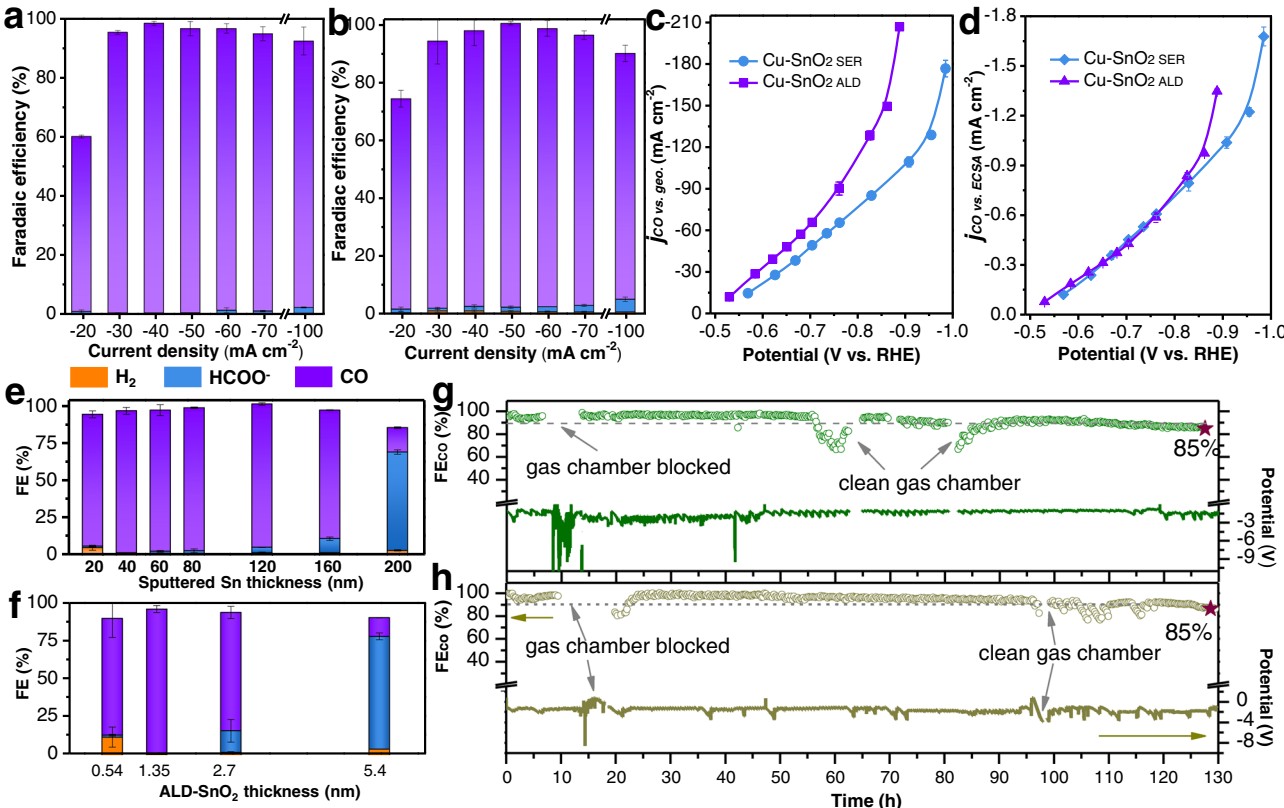

**Fig. 2 | Electrocatalytic performance of Cu-SnO₂ catalysts.** Faradaic efficiencies of products generated on (**a**) Cu-SnO₂ ALD and (**b**) Cu-SnO₂ SER catalysts under different current densities. Partial current densities of CO normalized against (**c**) geometric surface area and (**d**) electrochemical surface active area of two catalysts. Faradaic efficiencies of products produced on (**e**) Cu-SnO₂ SER and (**f**) Cu-SnO₂ ALD with different thickness of sputtered-Sn and ALD-SnO₂ overlayer under −50 mA cm⁻². Faradaic efficiency of produced CO and the detected half-cell potential under −40 mA cm⁻² over 130-h electrolysis on (**g**) Cu-SnO₂ SER and (**h**) Cu-SnO₂ ALD catalysts. Each data point in (**a**) to (**f**) corresponds to the average value from three to four independent measurements and the error bars represent the standard deviations of these measurements.

analyzed as a function of potential (Supplementary Fig. 21). Briefly, Cu-SnO₂ ALD displays the highest FE of CO of 98% at -0.63 V vs. RHE, with the FE maintaining at >90% from −0.59 to −0.77 V. As compared to Cu-SnO₂ ALD, Cu-SnO₂ SER requires 60 mV larger overpotential, −0.69 V vs. RHE to reach the best selectivity of 98% towards CO generation. At potential that is <-0.75 V, FE for H₂ and HCOO⁻ production starts to increase. C₂₊ products are largely suppressed on both catalysts, suggesting inhibition of further reduction of CO. Additionally, the pH of electrolyte and the nanostructure of Cu substrate have little impact on the maximum selectivity of CO on both catalysts (Supplementary Figs. 22–24).

The CO production rate is considerably improved on Cu-SnO₂ ALD and Cu-SnO₂ SER (Fig. 2c-d and Supplementary Fig. 25). Remarkable geometric partial current densities of −177 mA cm⁻² at −0.99 V vs. RHE and −207 mA cm⁻² at −0.89 V vs. RHE for CO production are achieved on Cu-SnO₂ SER and Cu-SnO₂ ALD, respectively (Fig. 2c). Specific current density normalized against electrochemical active surface area (Supplementary Table 3) shows little difference between two Cu-SnO₂ catalysts at potential > −0.9 V vs. RHE (Fig. 2d).

The effect of the sputtered Sn thickness on the catalytic performance of Cu-SnO₂ SER is investigated at a constant current density of −50 mA cm⁻² (Fig. 2e). By reducing the thickness of Sn from 60 nm to 20 nm, the production of CO remains dominant (Supplementary Fig. 26). Surprisingly, growing the thickness further to 160 nm still results in the selective formation of CO on Cu-SnO₂ SER catalyst (Fig. 2e), even though the Cu surface is mostly covered by nanoparticles that are mainly consisted of SnO₂ SER (Supplementary Fig. 27). Thus we assume that the SnO₂ SER could also be active for CO

generation in the presence of Cu. By contrast, the selectivity of Cu-SnO₂ ALD catalyst dramatically shifts to HCOO⁻ if the thickness increases from 2.7 to 5.4 nm (Fig. 2f, Supplementary Figs. 28)[27,28]. Note that both sputtered Sn and ALD-SnO₂ on GDE deliver a favorable HCOO⁻ formation in our reaction system (Supplementary Fig. 29).

Excellent stability of Cu-SnO₂ ALD and Cu-SnO₂ SER towards CO production is demonstrated by long-term electrolysis. With periodic removal of precipitated salt (Supplementary Fig. 30), both catalysts exhibit outstanding durability, with a steady FE of CO over 85% after -130 h at a current density of −40 mA cm⁻² and over 80% after -120 h at a current density of −100 mA cm⁻² (Fig. 2g-h, Supplementary Fig. 31).

## Structure-activity relationship as revealed by X-ray absorption spectroscopy

To interrogate the relationship between activity and the local structures of Cu-SnO₂ SER and Cu-SnO₂ ALD, we performed XAS at the Cu K-edge and Sn K-edge (Fig. 3, Supplementary Figs. 32–34). From the Cu K-edge, the conversion of CuO to metallic Cu after pre-treatment confirms the reduction process (Fig. 3a, b), consistent with our XRD analysis[19,29]. In contrast, the Sn K-edge results show that most Sn species in both catalysts retain a Sn(IV) feature even after pre-reduction of the catalysts (Fig. 3c, d), which is in a good agreement with the XPS analysis.

We then fitted the post-edges of the Sn K-edge XAS for both catalysts after reduction (Fig. 3e, Supplementary Fig. 34 and Table 13). On Cu-SnO₂ ALD, we find the presence of Sn-Cu bond in addition to Sn-O and Sn-O-Sn bonds. According to previous studies, Sn-Cu sites could contribute to the selective CO production[15]. However, these Sn-Cu

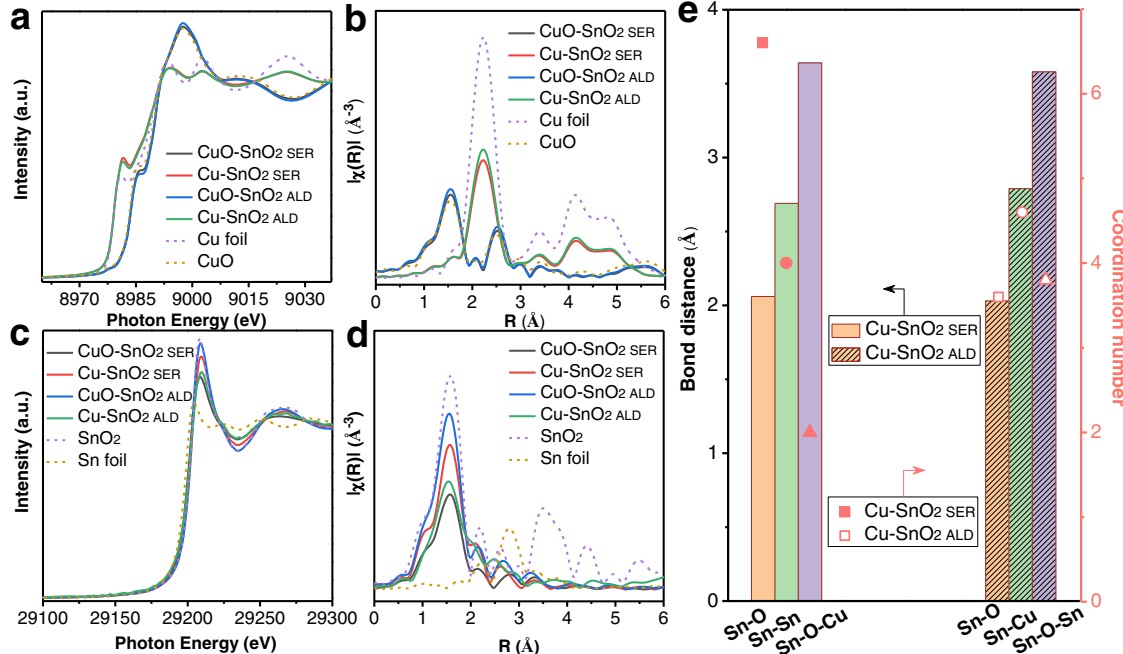

**Fig. 3 | X-ray absorption spectroscopy analysis.** The Cu K-edge XAS features of two as-prepared CuO-SnO₂ samples, two pre-reduced Cu-SnO₂ catalysts and two Cu standards (Cu foil and CuO) at the (**a**) near-edge and (**b**) post-edge after Fourier transform. The Sn K-edge XAS features of two as-prepared CuO-SnO₂ samples, two pre-reduced Cu-SnO₂ catalysts and two Sn standards (Sn foil and SnO₂) at the (**c**) near-edge and (**d**) post-edge after Fourier transform. **e** Fitting of the post-edge of Sn K-edge XAS of both Cu-SnO₂ catalysts after pre-reduction.

sites are not found in the Cu-SnO₂ SER. Instead, Sn-Sn bond formation is resolved in addition to Sn-O and Sn-O-Cu bonds. Taken together, while both Cu-rich and Sn-rich catalysts share a similar Cu-supported SnO₂ configuration, they hold different local structures for their respective production of CO from CO₂ reduction.

## Binding of key intermediates as probed by electrochemical CO adsorption and in situ Raman spectroscopy

Electrochemical cyclic voltammograms of three catalysts in both He and CO flowing conditions are shown in Fig. 4a and Supplementary Fig. 35[30]. The voltammograms have a slightly slanted baseline due to the reduction of the unavoidable residual oxygen molecules inside the pores of GDE substrate. Even though substitution of GDE by a rigid substrate, i.e. glass, renders a flat baseline, we choose to use GDE as the substrate for the following tests to ensure that the features represent the real catalytic condition in a flow cell. As compared to the CV curve of Cu, the peaks at 0.30–0.39 V on Cu-SnO₂ SER and −0.41--0.50 V on Cu-SnO₂ ALD originate from Sn/SnO₂ redox reactions (Fig. 4a and Supplementary Fig. 36)[13].

CO adsorption/stripping features were further characterized by cyclic voltammetry within a narrow potential window from −0.5 to 0.1 V vs. RHE under the flow of CO (Fig. 4b). The peaks at −0.35/-0.21 V and −0.24 V on Cu substrate are attributed to the CO stripping and adsorption on the Cu active site, respectively, as confirmed by the linearity between the cathodic peak current and the scan rate (Supplementary Figs. 37–38)[31]. These peaks are in excellent agreement with previous CO adsorption studies on polycrystalline Cu reported by Hori et al.[31,32].

The CO stripping peaks shift positively to −0.32 V and −0.16 V on Cu-SnO₂ ALD surface as compared to Cu, indicating the modified binding energy of CO on Cu sites with SnO₂ coating (Fig. 4b). Note that CO adsorption peak at −0.24 V becomes faint on Cu-SnO₂ ALD as compared to the one of Cu, due to the decreased amount of Cu sites with SnO₂ ALD overlayer. Surprisingly, a shoulder for the cathodic peak at −0.33 V and a broad anodic peak at −0.17 V, representing the CO

adsorption and stripping at the SnO₂ SER active site respectively, appear on the Cu-SnO₂ SER (Fig. 3b and Supplementary Fig. 39). The cathodic peak becomes more intense in the presence of Cu as compared to the non-modified sputtered Sn. These features provide evidence that the presence of Cu shifts the adsorption preference of Sn sites to C-bound *CO, thus altering the selectivity from HCOO⁻ on non-modified Sn to CO on Cu-SnO₂ SER catalyst. Based on the fact that the Sn-rich surface still shows remarkable CO production rate (Supplementary Table 1 and Fig. 2c-d), we believe that surface SnO₂ SER is the essential active site for CO production.

We also applied in situ Raman spectroscopy to investigate the adsorption of CO on three catalysts using a custom-made flow cell (Fig. 4c-e and Supplementary Figs. 40-42). The spectrum of Cu-SnO₂ SER catalyst is almost featureless since surface-enhanced Raman scattering (SERS) effect from Cu is largely inhibited by the thick Sn layer[18]. Cu and Cu-SnO₂ ALD show Raman peaks at wavenumbers of ~288, ~510 and ~604 cm⁻¹ at open circuit potential, representing oxides of Cu. When we change the current density from −20 to −100 mA cm⁻² on Cu, features associated with frustrated rotation of *CO, Cu-CO stretching and C≡O stretching appear at ~265, ~348 and ~2065 cm⁻¹, respectively[33]. Interestingly, Cu-SnO₂ ALD shows a blue-shift of *CO rotation peak from ~265 to ~284 cm⁻¹, indicating a higher excitation energy of the *CO restricted rotation on Cu active sites after coating SnO₂ overlayer. The upward shift of this rotational frequency is likely attributed to the electrochemical Stark effect[34], which is caused by the more negative potential of Cu-SnO₂ ALD required for selective CO formation as compared to Cu at the same current density (Supplementary Fig. 19).

We further scrutinized the Raman signal of the C≡O stretching mode at ~2000–2100 cm⁻¹ observed on Cu and Cu-SnO₂ ALD cathodes. In accordance with previous in-situ Raman investigations[35,36], we assign these peaks to the linearly adsorbed CO on atop Cu sites (*CO_atop). On Cu-SnO₂ ALD catalyst, the *CO_atop peak appears more intense as compared to *CO_atop on Cu at all current densities (Fig. 4d). As *CO_atop coverage is in dynamic equilibrium with local CO concentration[37,38],

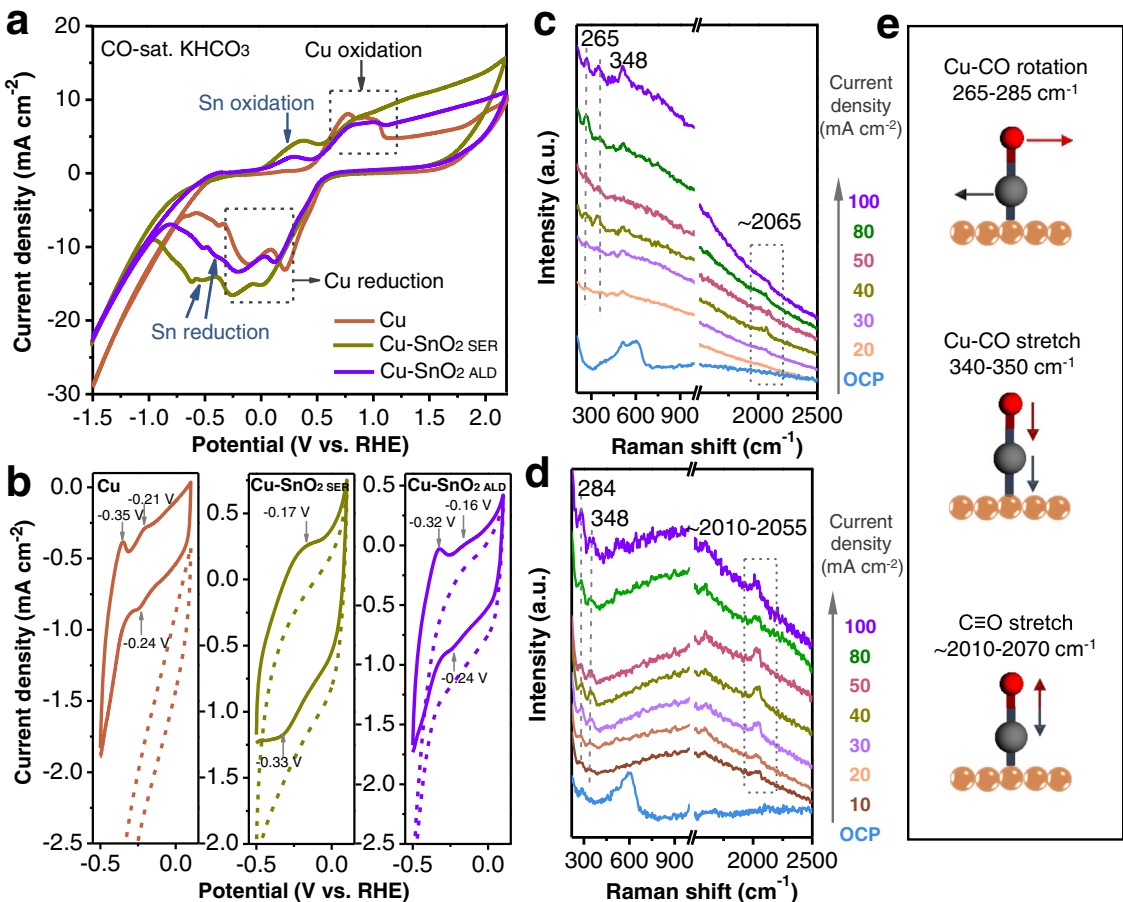

**Fig. 4 | CO adsorption feature and in situ Raman measurement on Cu and Cu-SnO₂ catalysts. a** Cyclic voltammograms of Cu, Cu-SnO₂ SER and Cu-SnO₂ ALD catalysts measured in a flow cell with 0.5 M KHCO₃ (pH = 8.36) and CO being flowed at 0.375 and 50 cm³ min⁻¹, respectively, at a scan rate of 50 mV s⁻¹. **b** Cyclic voltammograms of Cu, Cu-SnO₂ SER and Cu-SnO₂ ALD catalysts measured within a narrow potential window from −0.5 V to 0.1 V vs. RHE at a scan rate of 50 mV s⁻¹, with He (dash line) or CO (solid line) being flowed at 50 cm³ min⁻¹. In-situ Raman spectra of (**c**) Cu and (**d**) Cu-SnO₂ ALD catalysts during CO₂ reduction under different current densities. **e** Schematic view of three different modes of adsorbed *CO on a Cu active site.

the increase of peak intensity is consistent with the high current density of CO production on the Cu-SnO₂ ALD catalyst (Fig. 2a). In addition, the observed slight red-shift of the *CO$_{atop}$ peak is attributed to the weaker binding of CO on SnO₂-doped Cu defect sites[39–41].

The C≡O stretching band is further decoupled into two peaks with a low wavenumber (LW) of ~2010 and a high wavenumber (HW) of 2055 cm⁻¹ (Supplementary Fig. 41)[42]. It has been well-established that atop-bound CO on terraces produces a C≡O stretching band with ~30 cm⁻¹ lower Raman shift than CO adsorbed on undercoordinated defects sites[40,41]. A greater intensity ratio of CO$_{atop-LW}$/CO$_{atop-HW}$ observed on the Cu-SnO₂ ALD indicates that more CO is produced on terraces after coating the SnO₂ layer (Supplementary Fig. 41).

For comparison, we also carried out in situ Raman spectroscopic tests on Cu catalyst coated with 20 cycles ALD of SnO₂ (Cu-SnO₂ ALD20, Supplementary Fig. 42), which shows selective conversion of CO₂ to HCOO⁻ (Supplementary Fig. 28). Under bias, the features arising from the frustrated rotation of *CO, Cu-CO stretching and C≡O stretching appear at the same position, albeit with a weaker intensity of C≡O stretching peak as compared to Cu-SnO₂ ALD5. An additional broad band of C-H stretching mode appears at ~2900 cm⁻¹, which could be assigned to the adsorbed HCOO⁻ on Cu active sites[43,44]. Moreover, a small peak appears with a Raman shift of 510–515 cm⁻¹, which might be related to the hydroxyl intermediates adsorbed on Cu[45]. The hydroxyl species favors CO₂ adsorption in the form of H₂CO₃, which in turn will be transformed to HCOO* species, promoting the formation of formate[46,47]. In overall, the Raman features on Cu-SnO₂ ALD20 are

consistent with its high selectivity towards formate formation, indicating that SnO₂ overlayers with different thickness successfully tune the adsorption energy of different intermediates on the Cu active sites.

## Photosynthesis of CO from CO₂ on Cu-SnO₂ ALD catalyst

We developed an integrated PV-EC system to perform unbiased synthesis of CO from CO₂ and water, which provides a route to recycling CO₂ into a valuable chemical feedstock using renewable electricity.

$$2CO_2 \rightleftharpoons 2CO + O_2 \ (E^0 = 1.32\,V, \text{under ambient pressure and temperature})$$

Both FE$_{CO}$ and the cell voltage (V$_{cell}$) necessary to drive the uphill conversion are the key factors affecting the efficiency of electrolyzer as well as the overall solar to fuel (STF) energy conversion efficiency (Supplementary Note 3). We first designed a two-electrode flow cell based on Cu-SnO₂ ALD cathode for nearly exclusive production of CO (Fig. 5a and Supplementary Fig. 43). Further we optimized the electrolyzer including refining electrolyte by replacing KHCO₃ with KOH to decrease cathode overpotential, excluding the membrane and selecting electrodeposited IrO$_x$ as an appropriate anode (Supplementary Fig. 44), to minimize the cell voltage. As a result, an overall cell voltage of only ~2.0 V is required for maintaining a 95% faradaic efficiency for CO generation at a current density from ~40 to ~60 mA cm⁻²$_{cat}$ (current normalized by geometric surface area of cathode, 0.25 cm²).

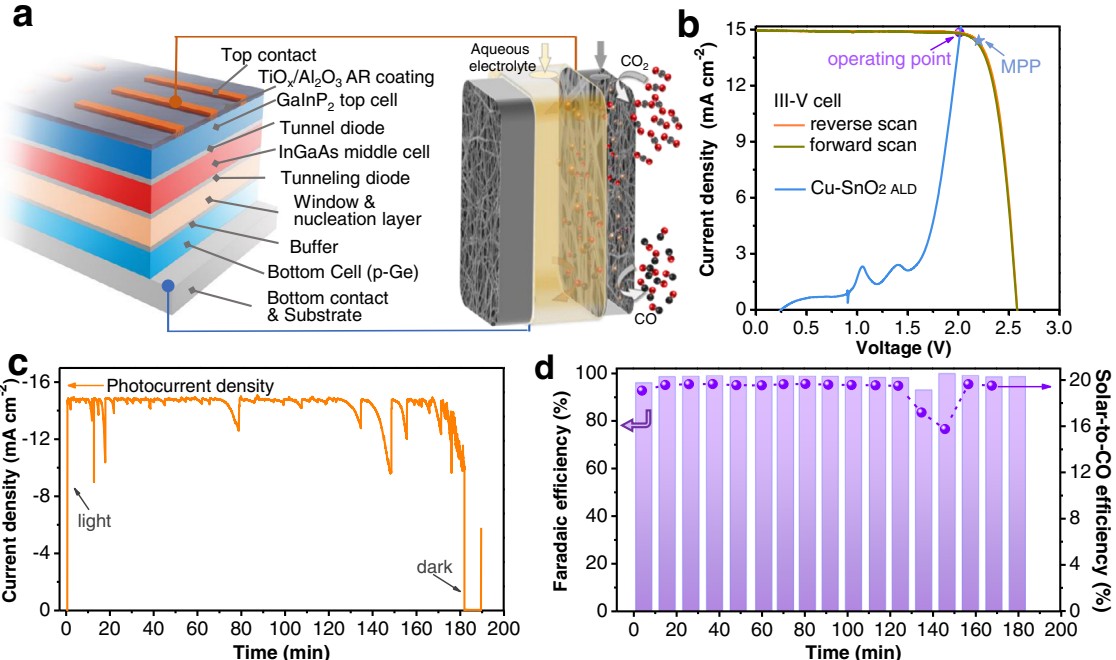

**Fig. 5 | Photosynthesis of CO from solar-driven CO$_2$ reduction. a** A schematic of the solar driven system which consists of a III-V triple junction solar cell and a flow reactor. **b** Representative linear sweep voltammograms of the solar cell and electrolysis cell. **c** Photocurrent density of the unassisted PV-EC system under standard AM 1.5 G illumination. **d** Faradaic efficiency of generated CO and calculated solar-to-CO conversion efficiency in the unassisted PV-EC system under standard AM 1.5 G illumination. The current density of electrolysis in (**b**) and (**c**) is normalized against working area of solar cell (0.92 cm$^2$). The performance of photovoltaic is measured under simulated standard AM 1.5 G with intensity of 100 mW cm$^{-2}$. MPP denotes the maximum power point of the solar cell.

On the basis of this electrolysis requirement, a III-V semiconductor, InGaP$_2$/InGaAs/Ge triple-junction photovoltaic cell was employed and directly connected with our flow electrolyzer via copper wire to provide the required cell voltage as well as necessary current for driving the CO$_2$ conversion to CO (Fig. 5a). The effective illumination area of the selected InGaP$_2$/InGaAs/Ge solar cell is 0.92 cm$^2$. Under simulated air mass 1.5 G irradiation, the triple-junction cell deliveres an open-circuit photovoltage (V$_{oc}$) of 2.57 V, a short-circuit current density of 14.96 mA cm$^{-2}$ and a fill factor of 82.0%, representing a power conversion efficiency (PCE) of 31.52% (Fig. 5b). By intersecting the *J-V* curves of the electrolyzer and the solar cell, we predict the operating point to be at 2.02 V and 14.83 mA cm$^{-2}$ (Fig. 5b). Note that here the electrolysis current is normalized against the effective illuminated area of the solar cell (0.92 cm$^2$). The unassisted PV-EC system delivers an average solar current density of 14.80 mA cm$^{-2}$ (Fig. 5c), with an average FE of 98.9% for CO production (Fig. 5d). This is a new benchmark for solar-to-CO conversion with a peak STF efficiency of 19.7% and an average value of 19.6% during 2-hr electrolysis (Fig. 5d, Supplementary Note 3). We note here that switching electrolyte from KHCO$_3$ to KOH is critical for achieving such a high conversion efficiency with little change of the binding of key intermediates (Supplementary Figs. 45-46). If the PV-EC system is performed in e.g. 0.5 M or 2 M KHCO$_3$, a current density of only -3.5 or -6.5 mA cm$^{-2}$ could be respectively achieved, leading to a STF efficiency of only 2.5% or 7.3%, respectively (Supplementary Fig. 46).

The CO production rate of our alkaline PV-EC configuration is calculated via the cathodic current density normalized against the surface area of Cu-SnO$_2$ $_{ALD}$ electrode (0.25 cm$^2$). The total current density of 54.5 mA cm$^{-2}_{cat}$ enables a CO production rate of 284.4 g h$^{-1}$ m$^{-2}$ (Supplementary Note 4), corresponding to a 200× increase over our previous work[7] and a 3× superior output compared to the state-of-the-art PV-EC CO$_2$ reduction system based on a noble metal cathode[12]. Accounting for the formation of H$_2$ and HCOO$^-$ by-products, a peak solar-to-fuel conversion efficiency of 19.9% is achieved. However, the deactivation of IrO$_x$ anode due to its dissolution leads to a significant drop of the current density after ~2 hr electrolysis (Supplementary Figs. 47-48). The dissolution of Sn is also observed (Supplementary Table 14). But neither the dissolved Ir nor Sn was re-deposited on their counterparts, as indicated by the absence of corresponding signals in the EDX spectrum (Supplementary Figs. 47-48). Further investigation on replacing IrO$_x$ by a stable and earth-abundant catalyst for oxygen evolution reaction is currently in progress.

With the deployment of gas-diffusion-electrode based flow cell, the production rate of CO in our system shows a 1 to 2 orders of magnitude improvement over previously reported systems using H-cell configuration (Supplementary Table 15). Besides an unprecedented solar-to-fuel energy conversion efficiency of our system, the usage of earth-abundant cathode as well as a 3 times enhancement of CO production rate over state of the art present a critical step towards further scale-up applications of this technology. Meanwhile, it is noted that strategies for long-term operation with affordable components of the integrated system are still required to make PV-EC system competitive with the fossil fuel derived alternatives.

In summary, earth-abundant electrocatalysts with metallic Sn as well as SnO$_2$ being introduced onto CuO nanowires are developed for efficient synthesis of CO from CO$_2$ reduction. In the custom-made flow cell with gas diffusion electrode supported catalysts, we achieve almost unit selectivity towards CO formation on both Cu-SnO$_2$ $_{SER}$ and Cu-SnO$_2$ $_{ALD}$ catalysts in 0.5 M KHCO$_3$ electrolyte. Partial current densities of -177 and -207 mA cm$^{-2}$ for CO generation are delivered by Cu-SnO$_2$ $_{SER}$ and Cu-SnO$_2$ $_{ALD}$ catalysts, respectively. On the basis of catalytic performance of Cu-SnO$_2$ $_{SER}$ with different Sn thickness as well as CO adsorption behavior on Cu-SnO$_2$ $_{SER}$ surface, we propose that the SnO$_2$ sites on Cu surface is selectively catalyzing CO generation. The intermediate adsorption configuration on the Cu-SnO$_2$ $_{ALD}$ catalyst observed by in situ Raman spectroscopy explains its excellent CO selectivity. Based on our highly efficient system for electrochemical

$CO_2$ reduction, a triple-junction $InGaP_2$/InGaAs/Ge photovoltaic is chosen to provide the needed energy for unassisted solar-driven $CO_2$ reduction. Direct coupling of solar device and flow electrolyzer enables a peak solar-to-CO conversion efficiency of 19.7% and average one of 19.6% during 2 hr-electrolysis under standard AM 1.5 G illumination. Our work serves as a guide for future design of artificial photosynthesis configuration, which is a promising strategy to produce commodity chemicals and fuels directly with only $CO_2$ and water as reactant and renewable energy as power source.

## Methods

### Preparation of electrodes

**Preparation of CuO nanowires substrate.** Polycrystalline Cu layer with a thickness of around 900 nm was first sputtered (DP650, Alliance-Concept) on the surface of gas diffusion electrode (GDE, Sigracet 38 BC, Fuel Cell Store). During sputtering, Cu target with high purity (99.99%) was bombarded under a direct current (DC, Huttinger TruPlasma 3002) power of 400 W with a sputtering rate of 2.65 nm $s^{-1}$. The pressure of the chamber was kept at $5.0 \times 10^{-2}$ mbar using both a dry pump (Pfeiffer-Adixen ACP 40) and a cryogenic pump (CTI ON-BOARD 8). The Cu/GDE substrate was then anodized in 3 M KOH (Reactolab, S.A.) electrolyte under a geometric current density of 8 mA $cm^{-2}$, till reaching a threshold voltage of 2.1 V. The anodization was carried out in a two-electrode setup, using a Cu foil (99.99%, Goodfellow) as the counter electrode. The resulting $Cu(OH)_2$ films were dried in air and annealed at 150 °C for 1 h under ambient atmosphere, leading to the formation of uniform CuO nanowires (NWs).

**Sputtering deposition of metallic Sn.** Metallic Sn with different expected thickness were sputtered on the surface of CuO NWs by DP650, with a deposition rate of 3.47 nm $s^{-1}$ and a DC power of 250 W.

**Atomic layer deposition of $SnO_2$.** The atomic layer deposition of $SnO_2$ was carried out in a Savannah 100 instrument (Cambridge Nanotech Inc.). Tetrakis(dimethylamino)-tin(IV) (TDMASn, 99.99% Sn, Strem Chemicals) was used as the precursor of Sn and DI water was used as the oxidizing agent, respectively. $N_2$ (99.9999%, Carbagas) was used as the carrier gas with a flow rate of 5 $cm^3$ $min^{-1}$. The temperature of TDMASn precursor and deposition chamber were respectively held at 60 °C and 110 °C during each deposition. Each cycle of $SnO_2$ deposition includes following steps: (a) pulsing TDMASn for 30 s, equilibrating for 15 s and purging for 60 s, (b) evacuating followed by pulsing DI water for 1 s, equilibrating for 15 s before purging for 20 s. This cycle was repeated for the desired number for preparing $SnO_2$ $_{ALD}$ overlayer with different thickness.

**Electrodeposition of $IrO_x$.** A sputtered Ti film with thickness of 30 nm onto GDE was used as the conductive substrate for the electrodeposition of $IrO_x$ film. The recipe of electrodeposition is based on our previous work[48]. Briefly, 200 mL aqueous electrolyte containing 0.409 g hexachloroiridic acid ($H_2IrCl_6$, ≥99.0%, Sigma-Aldrich), 0.714 g oxalic acid ($H_2C_2O_4$, ≥ 99.0%, Sigma-Aldrich) and 2 mL hydrogen peroxide ($H_2O_2$, 30%, Reactolab S.A.) was prepared and the pH value was adjusted to 10.5 by slowly adding $K_2CO_3$ (≥99.0%, Sigma-Aldrich). The as-prepared solution was left to stabilize for three days before usage. Cyclic voltammetry was used to deposit $IrO_x$ layer in a three-electrode setup where Ti/GDE, Pt foam and Ag/AgCl were used as the working electrode, counter electrode and reference electrode, respectively. The cylic voltammetry was performed within an applied potential window from 0.32 to 1.48 V vs. RHE at a scan rate of 1 mV $s^{-1}$ and 100 cycles of repeated scans were applied.

### Catalysts characterization

The surface morphology of the catalyst was characterized using a field emission scanning electron microscope (FE-SEM, Merlin, Zeiss), and elemental mappings were obtained from energy-dispersive X-ray (EDX) analysis. Cross-section of atomic layer deposited $SnO_2$ layer on planar CuO surface was measured by focus ion beam (FIB, Merlin, Zeiss). The samples were first protected by carbon film and then a hole was milled in front of it, creating a flat surface perpendicular to the sample surface. Transmission electron microscopy (TEM) was conducted on Talos, equipped with a high-angle annular dark field (HAADF) detector (FEI). The crystalline structure of the catalysts was analyzed by X-ray diffraction (XRD) on a Bragg-Brentano instrument coupled with a monochromated Cu Kα radiation (λ = 1.5409 Å) and a grazing incident beam. Diffractograms were recorded between 2θ = 10 to 70° with a slow scan rate of 0.25° $min^{-1}$ and a step size of 0.026°. The surface chemical compositions of the sample were measured by X-ray photoelectron spectroscopy (XPS) on a PHI VersaProbe II scanning microprobe (Physical Instrument AG, Germany). Analysis was carried out using a monochromatic Al Kα X-ray source of 24.8 W power. The Cu K- and Sn K-edges XAS spectra were recorded at the 20 BM beamline of the Advanced Photon Source in Argonne National Laboratory (Lemone, IL, US), in which fluorescence signals were collected. XAS data were processed with Demeter (v.0.9.26)[49]. For data analysis, we applied the same structure (Sn-doped $Cu_6SnO_8$) for the EXAFS fittings of both Cu-$SnO_2$ $_{ALD}$ and Cu-$SnO_2$ $_{SER}$ catalysts. Note that we have a pre-caution process of removing possible residual electrolyte ions on the electrode for all the samples after pre-treatment. Right after reduction, the catholyte chamber was infused with DI water to remove the residual ions to prevent the possible reaction between the ions with the catalyst. The sample was then removed from the cell. Both the front and back of the GDE were washed with DI water thoroughly for several times, followed by drying with air gun. For the XPS and XAS characterizations, the as-synthesized CuO-$SnO_2$ $_{SER}$ as well as all the samples after pre-treatment and 50-min reduction were immediately packed in vacuum after cleaning. The vacuum packaging helps to minimize the impact of atmospheric oxidation at the catalyst surface.

### Electrochemical measurements with a flow cell

**Electrochemical flow cell.** A custom-built flow cell was used for performing galvanostatic $CO_2$ reduction and conducting cyclic voltammetry. The cathodic and anodic chambers were separated by an anion exchange membrane (AEM, Fumasep FKS-50, Fumatech), or a cation exchange membrane (CEM, Fumasep FKE-50, Fumatech) when the catalyst delivers a high selectivity towards $HCOO^-$.

**Chronopotentiometric measurement and product quantification.** Aqueous $KHCO_3$ (99.999%, Sigma-Alrich) with a concentration of 0.5 M (pH = 8.36) or KOH (Reactolab, S.A.) with a concentration of 1.0 M (pH = 13.82) was pumped inside two electrolyte chambers using a peristaltic pump with a rate of 0.25 $cm^3$ $min^{-1}$. Gaseous $CO_2$ with high purity (99.998%, Garbagas) was flowed into cathodic and anodic chambers with flow rates of 20 and 5 $cm^3$ $min^{-1}$, respectively, controlled by mass flow controllers (MFC, Alicat Scientific). Three-electrode configuration was used for characterizing catalytic performance of Cu and both Cu-$SnO_2$ catalysts towards $CO_2$ reduction. A 200 nm-thick platinum film sputtered on the surface of GDE was used as the counter electrode (CE) and a KCl saturated Ag/AgCl (Pine) was used as the reference electrode (RE). The Ag/AgCl electrode was calibrated by a reversible hydrogen electrode (RHE, HydroFlex) before each-day measurement and the potential values used in this work are scaled to RHE scale as follows:

$$V_{RHE} = V_{Ag/AgCl(KCl\,sat.)} + 0.197V + 0.059V \times pH$$

All the chronopotentiometric (CP) measurements were conducted using a Gamry potentiostat (Interface 1000). The potential at each applied current density was determined by linear sweep voltammograms (LSV) measured on different catalysts. The voltage was recorded with *iR* drop being automatically corrected via a current interrupt mode. Each CP measurement employed a fresh working electrode and the electrolysis last 40-50 min. During each electrolysis, four gas aliquots were automatically sampled into an online gas chromatography (GC, Trace ULRTA, Thermo). A micropacked shin-carbon column (Restek) was used to separate the gas products and a pulse discharge detector (PDD, Vici) was used to quantify the products, coupled with a calibration curve obtained using standard gas mixtures (Carbagas) with all the reference gases ($H_2$, CO, $CH_4$, $C_2H_4$, $C_2H_6$ and $C_3H_6$). The electrolyte was collected during electrolysis and the dissolved liquid products were analyzed using high performance liquid chromatography (HPLC, Agilent Infinity II) after reaction. An organic acid analysis column (AMINEX HPX-87H) was used to separate the liquid products. A refractive index detector (RID) was used to quantify $HCOO^-$ and $CH_3COO^-$. A variable wavelength detector (VWD) was used to quantify the $C_2H_5OH$ and $C_3H_7OH$. Standard solution including HCOONa (for $HCOO^-$, > 99.0%, Fluka Analytical), $CH_3COONa$ (for $CH_3COO^-$, >99.0%, Sigma-Aldrich), $C_2H_5OH$ (≥99.8%, Fisher Scientific) and $C_3H_7OH$ (≥99.8%, Fisher Scientific) with known concentrations was used for calibration.

**CO adsorption on catalyst surface.** Aqueous 0.5 M $KHCO_3$ with pH of 8.36 was pumped into both catholyte and anolyte chambers at a rate of 0.25 $cm^3$ $min^{-1}$, and gaseous CO (99.998%, Carbags) or He (99.9999%, Carbags) was purged into cathodic gas chamber at a rate of 50 $cm^3$ $min^{-1}$. GDE-supported Pt and Ag/AgCl were used as the CE and RE, respectively. All the catalysts were pre-reduced at a constant current density of −30 mA $cm^{-2}$ for ~100 s before measurement. For each test, at least three cycles of cyclic voltammograms were conducted to ensure a steady state of the electrode surface. A potential window from −1.5 V to +2.2 V vs. RHE was chosen to measure all the oxidation and reduction peaks on Cu and both $Cu-SnO_2$ catalysts. Then the potential was carefully tuned from −0.6 V to +0.2 V vs. RHE to avoid the significant surface change, such as Cu oxidation, for the observation of the CO adsorption and stripping features on the surface active site.

**Pb underpotential deposition on Cu active site.** Evaluation of the surface Cu active sites was carried out by Pb underpotential deposition in an aqueous solution containing 0.1 M $HClO_4$ (70%, Merck) and 0.001 M $Pb(OAc)_2$ (≥99.99%, Sigma-Aldrich). The electrolyte and gaseous He (99.9999%, Carbags) were flowed into catholyte chamber and cathodic gas chamber at a rate of 0.25 $cm^3$ $min^{-1}$ and 20 $cm^3$ $min^{-1}$, respectively. GDE-supported Pt and Ag/AgCl were used as the CE and RE, respectively. All the catalysts were pre-reduced at a constant current density of −30 mA $cm^{-2}$ for ~100 s before measurement. The applied potential was first held at the initial potential, −0.21 V or -0.24 V vs. RHE, for 5 min before each scan, and voltammograms were recorded from −0.21 V to 0.29 V vs. RHE for Cu and $Cu-SnO_2$ ALD and −0.24 V to 0.19 V for $Cu-SnO_2$ SER. All the voltammograms were measured at a scan rate of 50 mV $s^{-1}$. The charge consumed for Pb stripping (oxidation) was determined by integrating the oxidation peak area and was used for analyzing the number of surface Cu active sites. Background curves were also measured using bare 0.1 M $HClO_4$ electrolyte under the identical condition. Each experiment was repeated using at least two fresh samples and an average of stripping charge was reported.

**Double layer capacitance measurement.** Electrochemical surface active area of Cu, $Cu-SnO_2$ SER and $Cu-SnO_2$ ALD was assessed by measuring their double layer capacitance. The experiments were carried out under almost identical condition as the one for performing $CO_2$ reduction, with 0.5 M $KHCO_3$ being flowed into catholyte chambers at a flow rate of 0.25 $cm^3$ $min^{-1}$ and He being flowed into gas compartments at a flow rate of 20 $cm^3$ $min^{-1}$, respectively. In order to establish the potential window without Faradaic current, the voltammograms were first scanned within a wide potential range from 0.35 to 0.85 V vs. RHE for Cu and 0.25 to 0.95 V vs. RHE for $Cu-SnO_2$ SER and $Cu-SnO_2$ ALD catalysts. Then the curves were collected within a non-faradaic current window at different scan rates of 20, 40, 60, 80 to 100 mV $s^{-1}$ for Cu and 10, 20, 40, 50, 60 mV $s^{-1}$ for $Cu-SnO_2$ SER and $Cu-SnO_2$ ALD catalysts.

**In situ Raman spectroscopy**

In situ Raman spectra were measured using a Horiba confocal Raman microscope equipped with a 638 nm-diode laser, a water immersion (WI) objective (100×, Olympus), a monochromator (600 grooves/mm grating) and a charge coupled device (CCD) detector. The WI objective was protected from the electrolyte by a Teflon film with thickness of 0.0005-inch (FEP, American Durafilm) and a layer of DI water between the objective and the protective film. The electrochemical $CO_2$ reduction was carried out in a home-made spectro-electrochemical flow cell, with aqueous 0.5 M $KHCO_3$ being purged into catholyte chamber at a rate of 0.25 $cm^3$ $min^{-1}$ and $CO_2$ being flowed into the cathodic gas chamber at a rate of 20 $cm^3$ $min^{-1}$, respectively. A biologic SP-200 potentiostat was used to apply a constant current density for pre-reduction as well as $CO_2$ reduction. For experiments performed on different catalysts, the electrode was first pre-reduced at -30 mA $cm^{-2}$, followed by the measurements under applying constant current densities. Each current density was applied for at least 5 min before the collection of the spectra to ensure a steady-state condition of the catalyst surface. All the spectra in the main text were shown as collected without baseline correction or any other post-treatment.

**Solar-driven $CO_2$ reduction**

**Electrolyzer.** The catalyst for measurement of solar-driven $CO_2$ reduction was prepared in the same way as described above. In order to optimize the CO production in aqueous 2 M KOH electrolyte, the number of ALD cycles of deposited $SnO_2$ was adjusted to 10. A film of electrodeposited $IrO_x$ was used as the counter electrode. A custom-built two-electrode flow electrolyzer was used for measuring linear sweep voltammetry (LSV) and conducting $CO_2$ reduction. The LSV curve was recorded in the applied potential from 1.0 to 2.5 V at a scan rate of 25 mV $s^{-1}$.

**Solar cell.** A triple-junction $InGaP_2$/InGaAs/Ge photovoltaic cell with a masked area of 0.92 $cm^2$ was measured under standard AM 1.5 G illumination using the Oriel LCS-100 Class ABB solar simulator (Newport). The measurement was carried out in air at room temperature. A certified silicon solar cell (Newport) was used to calibrate the light intensity before measurement. *J-V* characteristic of the device was assessed in both forward and backward directions within a potential range from 2.6 to 0 V with a scan rate of 25 mV $s^{-1}$ and a step time of 40 ms.

**Integrated device.** The solar cell was then wired to our two-electrode electrolyzer and the performance of the integrated PV-EC system was measured via chronoamperometry without applying bias under AM 1.5 G illumination. The produced gas products were periodically sampled into GC every ~11.25 min and the average current density during 1.5 min before each injection was used to quantify the faradaic efficiency of dominant products, $H_2$ and CO. The generated liquid products that dissolved in electrolyte was analyzed by HPLC after electrolysis.

**Reporting summary**

Further information on research design is available in the Nature Research Reporting Summary linked to this article.

## Data availability

The authors declare that all data supporting the results of this study are available within the paper and its supplementary information files or from the corresponding authors upon reasonable request.

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

## Acknowledgements

This work was financially funded by *Gebert Rüf Stiftung* under Microbials scheme 'Solar-Bio Fuels' with a grant number of GRS-080/19 (J.G and D.R.). This project also received funding from the European Union's Horizon 2020 research and innovation program under Grant Agreement 884444 (M.G.). This research used synchrotron resources of the Advanced Photon Source, an Office of Science User Facility operated for the US Department of Energy Office of Science by Argonne National Laboratory and was supported by the US Department of Energy under contract no. DE-AC02-06CH11357 and the Canadian Light Source and its funding partners (Y.Z.F.). J.L. acknowledges funding from the European Union's Horizon 2020 Research and Innovation program under the Marie Skłodowska-Curie Grant Agreement No. 838686. D.R. acknowledges the Young Talent Support Program from Xi'an Jiaotong University (HG6J002). The authors gratefully thank Pierre Mettraux (SCI, EPFL) for the XPS characterization and Natalia Gasilova for the measurement of ICP-OES.

## Author contributions

J.G. J.L. and Y.L contributed equally. J.G. and D.R. conceived the idea. J.G. designed and carried out all the electrochemistry experiments. D.R. performed TEM analysis and M.X. assisted in XRD characterization. J.L. and Y.F. helped with the XAS measurement and relative data analysis. Y.L. proposed the implementation of III-V solar cell to the electrolysis system for solar-driven $CO_2$ reduction. S.M.Z. coordinated the project. J.G., D.R. and M.G. supervised the project. J.G. and D.R wrote the manuscript which was finalized by all the other co-authors.

## Competing interests

The authors declare no competing interests.
