## [Peer Review File · Nature Communications]

Peer review comments,

First round review

Reviewer #1 (Remarks to the Author):

In this manuscript, Gao et al. used ALD and sputtering methods to prepare SnO₂ or Sn covered Cu nanowire catalysts, which are used in a gas-diffusion layer-based flow cell for highly selective CO₂-to-CO electrochemical reduction. In combination with a solar cell, the system can deliver 20% solar-to-CO energy conversion efficiency. Overall, suitable and necessary material characterizations were conducted, and the catalysts are of very high selectivity to CO. The solar-to-CO energy conversion efficiency is also top ones among literature reports. However, I still have many concerns that should be well addressed by the authors. Some technical issues and inconsistencies are listed at first:

1. The notations in Figure 1a are incorrect. "Cu/Sn or Cu/SnO₂" should be Sn/Cu or SnO₂/Cu.
2. In Figure 1b, why the GDL shows so many diffraction peaks, which quite affect more detailed analysis of XRD patterns of catalysts? Is that caused by metal contaminations? The XPS spectra of GDL should also be provided in SI.
3. Line 97: "Interestingly, high-angle angular dark field (HAADF) image and the energy dispersive X-ray photo spectroscopic (EDS) mapping of Sn/CuO show a mixture of Cu/CuO and SnO_x particles on the Sn/CuO surface (Supplementary Fig. 3), which further demonstrates the displacement reaction between Sn and CuO." It seems that the mapping of oxygen (and all elements overlapped mapping) is missing in Supplementary Fig.3, so it cannot tell the distribution of Cu/CuO and SnO_x. In addition, it seems that the distribution of Cu and Sn are partially overlapped in the particles on nanowires, whether Cu-Sn alloy phase is formed? From XAS analysis, there is no Cu-Sn bonds formed in this sample. So do EDS and XAS contradict?
4. Line 100: "As a contrast, 5-cycle ALD allows for SnO₂ coating with thickness of ~1.35 nm (Fig. 1e, Supplementary Figs. 4-5)." Where are the 5-cycle ALD data? I cannot see the thickness of SnO₂ coating.
5. Line 123: "No obvious change in Sn spectrum is detected for Sn/CuO, while the BE of Sn in SnO₂/CuO surface is 0.6 eV lower after pre-reduction". It is very clear that the Sn 3d_{5/2} of Sn/CuO also shifts to a lower BE after pre-reduction in Figure 1c, which has the same position to SnO₂/CuO after the pre-reduction. The reasons of peak shifts should also be briefly discussed.
6. In addition, in Figure 1c, the pre-reduction obviously increases the content of metallic Sn on the Sn/CuO catalyst according to XPS, probably due to reduction of spontaneously formed SnO₂ at negative potentials. In contrast, the SnO₂/CuO is still pure SnO₂. Is there any reason why the SnO₂ is so persistent in the SnO₂/CuO sample?
7. The Pb UPD curves in supplementary Fig. 9 are so weird. The cathodic scan should have more negative current densities in the presence of Pb²⁺ than those without Pb²⁺, and the anodic scan should have more positive current densities in the presence of Pb²⁺. The unstable air leakage (O₂) into the cell is possibly one reason.
8. Line 130: "Cu nanowires reconstructed into flakes after pre-reduction (Supplementary Fig. 10), resulting in a highly rough surface (Supplementary Fig. 11 and Supplementary Table 3)." From this Figure, I cannot see the flake-like structures. In addition, the curve recorded at 60 mV s⁻¹ in supplementary Figure 11c seems to have interference with quite periodic fluctuations.
9. Line 132: "Sn nanoparticles with size around 50 to 80 nm cover the surface of Cu NWs on Sn/Cu sample (Fig. 1f). SnO₂-based nanoparticles with the size of around 10 nm are dispersed on the surface of SnO₂/Cu sample (Fig. 1g)." From Fig. 1f and g, it is difficult to see the size of

surface nanoparticles. Herein, the referred Figures should be S7.

10. High-resolution HAADF-EDX characterizations (similar to Fig. S3) are also needed for SnO/CuO.

11. Line 140: "Additional characterizations of three catalysts after 50 min electrolysis at -50 mA cm⁻² are shown in Supplementary Fig. 13 and Table 4." From the SEM-EDS, the Sn contents seem to decrease obviously, can the authors make some comments and discussions?

12. In Table S5, the total FEs of Cu are significantly deviated from 100% from 20 to 100 mA cm⁻² current density, which are not observed on Sn/Cu and SnO₂/Cu. Why did it happen?

13. Similar to Pb UPD, the CO adsorption/stripping features are also weird. If the authors look into the referred works by Hori, they can find the CV with or without CO should share similar base lines, and the curves should not be so tilted.

14. I do not agree with the discussions in line 289 to 303. The two *CO bands above 2000 cm⁻¹ are both CO_{atop} without any doubt. *CO-bridge has much lower wavenumbers than CO_{atop} (e.g., 1800-1900 cm⁻¹ in neutral pH, J. Am. Chem. Soc. 2020, 142, 6, 2857-2867; ACS Catal. 2018, 8, 8, 7507-7516). The presence of two close CO_{atop} peaks are likely originated from different structures of adsorption sites (J. Phys. Chem. C 2017, 121, 22, 12337-12344; 2019, 9, 1, 474-478).

Then, I also have additional concerns regarding the novelty and significance of this work to the CO₂ reduction community. At first, some highly related works with thin SnO₂ layer on Cu and of high CO selectivity have been reported previously, but they are not cited and discussed in this work (e.g., J. Am. Chem. Soc. 2017, 139, 12, 4290-4293; ACS Catal. 2016, 6, 5, 2842-2851). Although they are not tested in flow cell and are not connected to solar cell, the authors should stress the novelty of this work and what new insights are provided in the material design part.

Then, the solar-to-CO efficiency is quite high in this work. I am curious whether such amazing efficiency is mainly related to the high efficiency of the solar cell or system integration? It should be mentioned that CO₂-to-CO conversion at low overpotentials, with high current densities, and with over 95% FE has been achieved in many catalysts.

In addition, since the authors are using a GDL flow cell, the current density used for long-term stability test is too small, only 40 mA cm⁻², which can even be achieved in H-type cell. Researchers are more interested in seeing the stability data at hundreds of mA cm⁻² operation in flow cell (e.g., Nature Communications volume 11, Article number: 593 (2020); Nature Energy volume 5, pages1032-1042 (2020); Nature Energy volume 5, pages684-692 (2020)).

Besides, regarding Sn/CuO prepared by sputtering, the surface Sn nanoparticles are of 50-80 nm in size. Hence, it is expected that there are no interactions between the surface Sn nanoparticles and Cu nanowire substrate as strain and ligand effects usually diminish within 10 nm. And XAS characterizations show no Sn-Cu bonds. I am curious what is the role of Cu in this catalyst? Whether Sn nanoparticles of similar size also selectively produce CO in this reactor?

Finally, the Sn/Cu and SnO₂/Cu catalysts have distinct structures and composition, but their CO selectivity and current densities are similar. In combination with the technical issues I mentioned above, overall, I feel the relationship between the catalyst structures and performance is unclear in this work.

Reviewer #2 (Remarks to the Author):

Comments on the manuscript entitled "Solar reduction of carbon dioxide on copper-tin electrocatalysts with energy conversion efficiency near 20%" by Jing Gao et. al.

Comments:

The authors reported that sputtered Sn-modified CuO nanowires (Sn/CuO) for production of CO in flow electrolyzer. Sn-rich Sn/CuO catalysts evidenced by X-ray absorption (XAS) and X-ray photoelectron spectroscopy (XPS) achieved FE_{CO} > 98%, j_{CO} ~ 177 mA/cm², and a stability of > 130 hours for CO production. The selectivity of Sn/CuO shifts from CO to formate upon an increase of surface Sn thickness, resulting from increased binding strength of *OCHO rather than *COOH. The authors also showed the solar-driven CO₂ reduction to CO with InGaP₂/InGaAs/Ge triple-junction solar cell system on Sn/Cu catalysts under simulated air mass 1.5G standard illumination, and achieved energy conversion efficiency near 20%. The idea of this work is quite interesting, but such an idea on CO₂RR catalysts employing the electrochemical cells powered by photovoltaics (PV-EC) system has already been reported, as in "ACS Energy Lett., 2020, 5, 470–476", and "J. Am. Chem. Soc., 2020, 142, 6878-6883." The difference seems to be only non-noble Cu-Sn materials instead of noble metals in the previous work. This work is evaluated to be lacked originality in publishing to Nature Communications. I think that this paper is better to transfer to other specialized journals and the comments in below will be needed further consideration for publication.

1) In Figure 1, it showed that only Sn (200), SnO₂ (200), and Cu (111) plane in TEM images while the various peaks of Sn, Cu, and even CuO are observed in the XRD results. Also, the author needs to provide SAED patterns of Sn-Cu catalysts.

2) In Figure 2, total faradaic efficiency (FE_{total}) of Sn/Cu catalysts showed less than 100% in all ranges of current density scale from 20 to 100 mA/cm². Especially, the FE_{total} of Cu showed around 70%, and thereby the author needs to clearly confirm the results of faradaic efficiency of all products.

3) In Figure 2, the author changed the Sn-Cu catalysts during the stability tests because the liquid flooding and the resulting salt precipitations degraded the catalytic properties. In Figure 2g and h, the potential results seem to be unstable during the reaction and how often do you change the Sn-Cu catalysts during the stability tests?

4) In this paper, the result of faradaic efficiency (FEs) on Sn-Cu catalysts was only plotted as a function of current density. To compare the effect of Sn on Cu surface more clearly, I think the results of FEs as a function of applied potentials are recommended.

5) In Figure 3, the author claimed that oxygen from the inner CuO layer on Sn/CuO catalysts migrates toward the outer Sn layer during the Cu reduction, resulting in the formation of metallic Cu-supported SnO₂ structure by XAS results. I think the author needs to explain more clearly and precisely in this results.

6) In Figure 4, the results of in-situ Raman showed that the Sn/Cu catalysts showed additional peaks of CO stretch (2010 ~ 2055 cm⁻¹) compared to Cu catalysts during the reaction. However, they showed similar Cu-CO rotation and stretch peaks. Please explain more details on the effect of Sn and how different results depend on the thickness of Sn layers.

Reviewer #3 (Remarks to the Author):

The article by Gao and Li et al., explores Sn incorporation on the copper oxide support to evaluate its ability for CO₂ to CO conversion. The authors have further extended this work to plug in a solar cell to the electrolyzer to demonstrate direct solar-to-fuel conversion ability of the catalyst. The results are interesting and I recommend it to be published after minor revision and answering the following questions and comments:

1. The introduction has appropriately covered the background however, there is still a need to briefly present, which material is the current state-of-art in the conditions explored by the authors. This can span materials used for PV-EC for CO₂ to CO conversion and also the ones, which show the potential from single cell studies for completeness.

2. The active catalyst explored in the present article is also the catalyst which was reported as active and selective in their previous work (Reference 16). Why is the overall outcome in the respective PV-EC setup different? Is it only due to pairing with a different anode and electrolyte? Comment on it and also include it explicitly in the text with explanations.
3. How do the authors exclude the presence of hydroxylated Sn and Cu species, especially after pre-reduction of CO₂RR? This aspect needs to be addressed in the text or else the possibility should be mentioned along with its plausible influences.
4. How do the authors envision the mechanism of spontaneous displacement between Sn and CuO mentioned? Is it self-limiting or extends to the near surface region or any other scenario?
5. What is the peak in XRD of as-prepared sample around same 2θ as for Cu(111) after pre-reduction shown in Figure 1b?
6. Does the sputter deposition of Sn allow for intermixing of Sn and Cu rather than just coating the top surface? Can this aspect be the reason for catalyst performance versus previous reports mentioned in line 196-198?
7. Are the XAS studies performed in situ or ex situ?
8. The fitting of post-edge XAS data points towards the presence of Sn-Sn bond for one case and Sn-Cu for the other. Since different components are considered for the fitting of the two catalysts, how does the fit turn out considering similar components?
9. Given XAS shows the presence of different sites on the two catalysts explored, why is the selectivity for both similar towards CO₂RR (line: 231-234)? Typically, the presence of Sn-Sn should result in more formate.
10. CVs in Figure 4b and S32c (add label a,b,c in the figure) for Sn/Cu shows a capacitive behavior perhaps due to rough vs. near-planar catalyst. How to decouple the effect of CO₂ being trapped aiding intermixed Sn-Cu to allow further reduction versus more uniform SnO₂ from ALD? Besides, the authors point out about the CO adsorption on Sn being intense in the presence of Cu (line: 264-266) but the desorption peak on Sn or SnO₂ is quite intense compared to Sn/Cu, which cannot be the case unless they also adsorb CO. Comment on it.
11. The catalyst characterization has been carried out in 0.5 M KHCO₃ throughout the entire work but the final PV-EC application is studied in 2M KOH. How can the understanding of the catalyst in a near neutral media be directly used to explain the outcome of functioning in a highly basic electrolyte? How is the catalyst performance in the PV-EC configuration using 0.5 M KHCO₃ instead?
12. XPS survey scan of the two catalysts coated on GDE should be provided to confirm absence of any other contributions.
13. What are the additional peaks in XRD of Fig. S1(a & b) for GDE? Peaks for CuO and Sn are barely visible, zoomed insets or separate figures are recommended along with goodness of fit for the claims. An alternative can be catalyst deposition on another planar substrate (glassy carbon for instance) to confirm the claims by eliminating GDE influence. Are XRD peaks for Sn and SnO₂ are expected at the same 2θ ? Fig. S1c, a Shirley background is used for SnO₂/CuO and linear for the other two cases, comment on it. Additionally, binding energy scales for XPS is typically represented from higher to lower values on the x-axis.
14. Fig. S7(f): Zoomed TEM showing SnO₂ layer around 1.3 nm is recommended to support the outcome from Fig. S5.
15. Fig. S9: Same x-y scaling is recommended for better comparison, provide zoomed insets or separate figures wherever necessary. What is the reason for slanted background in the CVs?

16. Fig. S11: Electrolyte used for the measurement is not mentioned. Also, CV in a wider potential ranged should be shown to justify no/minimal interference from faradaic currents. What is the reason for noisy traces in Fig. S11(c)? Also, what are the redox features observable in Fig. S11(b) for higher scan rates? How was the roughness factor computed? What value did the authors use for a flat surface and why?

17. Amount of Sn is shown to decrease after pre-reduction (Fig. S12 vs. S13), what is the expected cause? Does it go into the electrolyte? Could that be the reason that eventually more Cu is exposed giving the observed activity?

18. Fig. S37: What is the peak around 3.3 eV? Does Sn deposit over the anode?

19. Fig. S38: Significant Sn is lost during the 3h electrolysis (compared to Fig. S6 & S12). This should be explicitly mentioned in the text with the possible causes.

20. SI Table S1: How did the authors ensure removal of K from the catalyst after pre-reduction and before XPS analysis?

21. SI Table S2: Mention the background assumed for charge integration. Comment on the interaction of Pb and Sn during Pb-UPD.

22. SI Table S4: SnO₂/Cu catalyst shows no Sn detected in ICP but EDS shows a decrease in Sn content from 2.6% to 1.5% post electrolysis. Comment on it.

23. Revise the paper for minor typos like: (a) Line 79: X-ray diffraction (vs. X-ray diffractive); (b) Line:243-244 not clear, rephrasing will be better; (c) denotation of the catalyst is many times reversed which is confusing, make it consistent. Examples: Fig 4a (Cu/OD Sn); Fig 4b (Cu/Sn); Fig 5b (Cu/OD Sn); Fig. S7 (CuO/Sn); Fig. S16 (Cu/SnO₂); Fig. S18a; (d) Line 283: When instead of with the reduction current...; (e) Line 300: coating (vs. coting); (f) Line 314: PV-EC (vs. PV-FC); (g) Line 555: quantify (vs. qualify); (h) SI Line 54-55: Efuel (vs. Efule); (h) SI Line 93-94: Incorrect label description for Fig. S4.; (i) Represent either by EDS or EDX uniformly throughout the text; (j) SI Line 186-188: Meaning not clear, rephrase; (k) Fig. S29: He-trace barely visible; (l) Several instances in the text show different representation of the triple-junction solar cell (for example: Fig. S35 image & caption; SI Table S13) is that intentional, were different configurations used?; (m) SI Line 326: Scale (vs. Scar); (n) Fig. 5b: MPP is not explained anywhere in the text; (o) Line 378: strategy to produce (vs. strategy produce).

Our point-by-point response to the referees' comments is listed below in blue italic fonts.

Note: The as-prepared samples are now named as CuO-SnO₂_{SER} and CuO-SnO₂_{ALD}, where SER and ALD mean spontaneous exchange reaction and atomic layer deposition, respectively. After pre-reduction, the two samples were named then as Cu-SnO₂_{SER} and Cu-SnO₂_{ALD}.

Reviewer #1 (Remarks to the Author):

In this manuscript, Gao et al. used ALD and sputtering methods to prepare SnO₂ or Sn covered Cu nanowire catalysts, which are used in a gas-diffusion layer-based flow cell for highly selective CO₂-to-CO electrochemical reduction. In combination with a solar cell, the system can deliver 20% solar-to-CO energy conversion efficiency. Overall, suitable and necessary material characterizations were conducted, and the catalysts are of very high selectivity to CO. The solar-to-CO energy conversion efficiency is also top ones among literature reports. However, I still many concerns that should be well addressed by the authors. Some technical issues and inconsistencies are listed at first:

We thank the referee for the appreciation of our work. We have now addressed the remaining technical issues and inconsistencies in the revised manuscript.

1. The notations in Figure 1a are incorrect. "Cu/Sn or Cu/SnO₂" should be Sn/Cu or SnO₂/Cu.

We have corrected the notation in the revised manuscript.

2. In Figure 1b, why the GDL shows so many diffraction peaks, which quite affect more detailed analysis of XRD patterns of catalysts? Is that caused by metal contaminations? The XPS spectra of GDL should also be provided in SI.

The commercial GDL used in this work is based on carbon paper covered by a hydrophobic polytetrafluoroethylene (PTFE) layer (<https://www.fuelcellstore.com/sigracet-38-bc>). The peaks shown in the diffractogram of GDL are now assigned to carbon (PDF #00-026-1076), (C₂F₄)_n (PDF #00-061-1414) and CF₄ (PDF #00-055-0072) species from PTFE layer, as indicated by the reference added in the revised Supplementary Figure 1. Moreover, we also prepared the catalysts onto glass substrate. The XRD patterns clearly show Sn(200), Sn(101), Sn(211), CuO(002), Cu(111) and Cu(200) for the as-prepared samples and Sn(200), Sn(101), Cu(111) and Cu(200) for the pre-reduced catalysts. This result demonstrates that no metal contaminant is present in our catalyst. Following the reviewer's comment, we have also provided the XPS spectra of bare GDL in the Supplementary Figure 2 in the revised Supplementary Information. New discussions are now included from Line 72-80.

3. Line 97: "Interestingly, high-angle angular dark field (HAADF) image and the energy dispersive X-ray photo spectroscopic (EDS) mapping of Sn/CuO show a mixture of Cu/CuO and SnO_x particles on the Sn/CuO surface (Supplementary Fig. 3), which further demonstrates the displacement reaction between Sn and CuO." It seems that the mapping of oxygen (and all elements overlapped mapping) is missing in Supplementary Fig.3, so it cannot tell the distribution of Cu/CuO and SnO_x. In addition, it seems that the distribution of Cu and Sn are partially overlapped in the particles on nanowires, whether Cu-Sn alloy phase is formed? From XAS analysis, there is no Cu-Sn bonds formed in this sample. So do EDS and XAS contradict?

The additional mapping of elements is now added (Figure R1). The oxygen distribution around Sn particles is also reported. Since EDX mapping is a 2D view of 3D particles/nanowires, the overlapping between Sn and Cu in the 2D image is unavoidable even if Sn and Cu are phase-separated. Thus, other

methods such as XRD and XAS provide more precise information on whether Cu and Sn are phased-mixed. XRD pattern of the as-prepared Cu-SnO₂_{SER} catalysts show no peak that can be ascribed to alloy (Figure 1b). Thus, we believe that there is no observable interphase mixing between Cu and Sn in our Cu-SnO₂_{SER} sample. Revised text is shown in Line 113-116. New data is now included as Supplementary Figure 4.

Figure R1. (a) Representative high-angle annular dark field image of the as-prepared CuO-SnO₂_{SER} with a thickness of 60 nm for Sn overlayer, respective energy dispersive X-ray spectroscopic mapping of (b) Cu, (c) Sn, (d) O, (e) overlapped Cu, Sn and O. Scale bars: 50 nm for a to e.

4. Line 100: “As a contrast, 5-cycle ALD allows for SnO₂ coating with thickness of ~1.35 nm (Fig. 1e, Supplementary Figs. 4-5).” Where are the 5-cycle ALD data? I cannot see the thickness of SnO₂ coating.

The SEM image of as-prepared 5-cycle-ALD-SnO₂ coated Cu is shown in Figure 1e. However, the ultrathin layer of SnO₂ is not discernible in SEM. The HAADF-EDX mapping of Cu, O and Sn demonstrate the even distribution of Sn component on the surface of CuO nanowires (Figure R2). However, it is still challenging to determine the exact thickness of SnO₂ layer via HAADF-EDX. Thus the thickness of SnO₂ is determined via a linear fitting between thickness and deposition cycles via equation:

$$d_{\text{thickness}} = N * d_0 \text{ (N represents the number of cycles, } d_0 \text{ represents the thickness per cycle)}$$

The thickness of SnO₂ layer with different deposition cycles (from 200 to 400) was measured by the cross section of planar-CuO-supported SnO₂ using focus ion beam (Figure R3). According to the calibration curve of the thickness as a function of deposition cycles, the thickness of 5-ALD-cycle SnO₂ film was estimated to be 1.35 nm. New data are included as Figure 1e and Supplementary Figures 5, 8, 9

Figure R2. (a) Representative high-angle annular dark field image of the as-prepared CuO-SnO_2 ALD with a thickness of ~ 1.35 nm for SnO_2 overlayer; respective energy dispersive X-ray spectroscopic mapping of (b) Cu, (c) Sn, (d) O, (e) overlapped Cu, Sn and O. Scale bars: 50 nm for a to e.

Figure R3. FIB cross-section view of the planar CuO supported ALD- SnO_2 layer with different deposition cycles: (a and d) 200 cycles, (b and e) 300 cycles and (c and f) 400 cycles. Scale bars: $2 \mu\text{m}$ for a to c and 400 nm for d to f.

5. Line 123: “No obvious change in Sn spectrum is detected for Sn/CuO , while the BE of Sn in SnO_2/CuO surface is 0.6 eV lower after pre-reduction”. It is very clear that the Sn $3d_{5/2}$ of Sn/CuO also shifts to a lower BE after pre-reduction in Figure 1c, which has the same position to SnO_2/CuO after the pre-reduction. The reasons of peak shifts should also be briefly discussed.

Both Cu and Sn are easily oxidized during air exposure due to their oxophilicity. In order to avoid the impact of atmospheric oxidation on the surface composition, the as-synthesized CuO-SnO_2 SER and three samples after pre-reduction were immediately packed in vacuum after the sputtering or pre-reduction process. All the XPS spectra are now shown in Figure R4.

As compared to Sn 3d signals of as-synthesized SnO_2/CuO , a binding energy shift from 486.7 to

486.3 eV was still detected in Sn spectrum of the catalyst after pre-reduction. This result is likely due to the partial reduction of Sn^{4+} to Sn^{2+} in the presence of Cu, which is also observed by Pardo Pérez et al in their Cu nanowires supported SnO_2 catalyst (*Adv. Energy Mater.* 2022, 12, 2103328). New data is now included as Figure 1c. New discussions are now included from Line 117-127.

Figure R4. XPS of as-prepared $\text{CuO-SnO}_2_{\text{SER}}$, $\text{CuO-SnO}_2_{\text{ALD}}$ and the ones after pre-reduction at -30 mA cm^{-2} for $\sim 100 \text{ s}$.

6. In addition, in Figure 1c, the pre-reduction obviously increases the content of metallic Sn on the Sn/CuO catalyst according to XPS, probably due to reduction of spontaneously formed SnO_2 at negative potentials. In contrast, the SnO_2/CuO is still pure SnO_2 . Is there any reason why the SnO_2 is so persistent in the SnO_2/CuO sample?

The persistence of SnO_x species has also been reported in several previous reports such as J. Am. Chem. Soc. 2012, 134, 4, 1986-1989, J. Am. Chem. Soc. 2017, 139, 12, 4290-4293; Nat. Energy 2017, 2, 17087; Adv. Energy Mater. 2022, 12, 2103328. Recently, Mayer and co-workers show that this persistency of SnO_2 is critical for the selective formation of CO in Cu- SnO_2 catalysts. The persistency may be caused by the partial electron transfer from Sn to Cu species (ACS Catal. 2019, 9, 10, 9411-9417).

The comparison between our Cu/ $\text{SnO}_2_{\text{SER}}$ and Cu/ $\text{SnO}_2_{\text{ALD}}$ shows that the latter has a more persistent SnO_2 after reduction (Figure R4). This is probably caused by the structural difference between two samples. In Cu/ $\text{SnO}_2_{\text{ALD}}$, SnO_2 is an ultra-thin layer. In Cu/ $\text{SnO}_2_{\text{SER}}$, SnO_2 exists as large particles. The 'electron transfer effect' may not be effective for large SnO_2 particles, resulting in the existence of metallic Sn species in Cu/ $\text{SnO}_2_{\text{SER}}$. New data is now included as Figure 1c.

7. The Pb UPD curves in supplementary Fig. 9 are so weird. The cathodic scan should have more negative current densities in the presence of Pb^{2+} than those without Pb^{2+} , and the anodic scan should have more positive current densities in the presence of Pb^{2+} . The unstable air leakage (O_2) into the cell is possibly one reason.

We agree with the reviewer that the presence of oxygen could largely affect the shape of CV scans.

Thus, we re-performed the Pb UPD using the electrolyte that was saturated with He. He gas was continuously purged into the sealed flow cell during each Pb UPD test. The updated cyclic voltammograms are now provided in Figure R5.

Figure R5. Representative cyclic voltammograms showing the stripping and underpotential deposition of Pb on Cu active sites from (a) Cu nanowires, (b) Cu-SnO₂_{SER} and (c) Cu-SnO₂_{ALD} catalysts using the electrolyte containing 0.1 M HClO₄ and 0.001 M Pb(OAc)₂, the scan rate is 50 mV s⁻¹ for all the voltammogram tests.

Though the cyclic voltammograms are slightly tilted due to the unavoidable residual oxygen within gas diffusion layer, the peaks are clearly discernible for three different samples. As shown in Figure R6, catalyst deposited on a rigid FTO substrate shows normal cyclic voltammogram. New data is included as supplementary Figure 12. See revised text from Line 237-241.

Figure R6. Representative cyclic voltammogram of CuO deposited on an etched FTO substrate, showing the flat baseline.

8. Line 130: “Cu nanowires reconstructed into flakes after pre-reduction (Supplementary Fig. 10), resulting in a highly rough surface (Supplementary Fig. 11 and Supplementary Table 3).” From this Figure, I cannot see the flake-like structures. In addition, the curve recorded at 60 mV s⁻¹ in supplementary Figure 11c seems to have interference with quite periodic fluctuations.

We have updated the SEM image of Cu nanowires, Cu-SnO₂_{SER} and Cu-SnO₂_{ALD} with appropriate magnification (Figure R7). As compared to the as-prepared CuO nanowires, it is clearly seen that

reduced metallic Cu nanowires split into several flakes that consist of small nanoparticles.

Figure R7. Representative SEM images of (d) Cu, (e) $\text{Cu-SnO}_2_{\text{SER}}$ and (f) $\text{Cu-SnO}_2_{\text{ALD}}$ catalysts after pre-reduction at -30 mA cm^{-2} for $\sim 100 \text{ s}$. Scale bars: 200 nm for c to f.

The periodic fluctuations that appear on the CV curves might be due to electromagnetic interference from the external environment. We have then updated the CV measurements on two catalysts as well as Cu substrate (Figure R8). New data are included as Supplementary Figure 11, 13 and Supplementary Table 3.

Figure R8. Representative cyclic voltammograms on (a) Cu, (c) $\text{Cu-SnO}_2_{\text{SER}}$ and (e) $\text{Cu-SnO}_2_{\text{ALD}}$ catalysts measured from 0.35 to 0.85 V vs. RHE for Cu and 0.25 to 0.95 V vs. RHE for $\text{Cu-SnO}_2_{\text{SER}}$ and $\text{Cu-SnO}_2_{\text{ALD}}$ catalysts. Representative cyclic voltammograms on (b) Cu, (d) $\text{Cu-SnO}_2_{\text{SER}}$ and (f) $\text{Cu-SnO}_2_{\text{ALD}}$ catalysts within a non-faradaic current window.

9. Line 132: “Sn nanoparticles with size around 50 to 80 nm cover the surface of Cu NWs on Sn/Cu sample (Fig. 1f). SnO_2 -based nanoparticles with the size of around 10 nm are dispersed on the surface of SnO_2/Cu sample (Fig. 1g).” From Fig. 1f and g, it is difficult to see the size of surface nanoparticles. Herein, the referred Figures should be S7.

We have updated Figure 1 in the revised manuscript for a clear demonstration of the morphology of both catalysts. The distribution of quite small SnO_2 nanoparticles, are now visible on the surface of

Cu nanowires (Figure R7e). HAADF and corresponding EDX mapping images of the pre-reduced catalysts are included in Supplementary Figure 16. Revisions are made from Line 134 to 138.

10. High-resolution HAADF-EDX characterizations (similar to Fig. S3) are also needed for SnO₂/CuO.

HAADF-EDX characterizations were performed for CuO-SnO₂ ALD (Figure R2). The new data is included in the Supplementary Figure 5.

11. Line 140: “Additional characterizations of three catalysts after 50 min electrolysis at -50 mA cm⁻² are shown in Supplementary Fig. 13 and Table 4.” From the SEM-EDS, the Sn contents seem to decrease obviously, can the authors make some comments and discussions?

We thank the referee for pointing out this important phenomenon. For Cu-SnO₂ SER catalyst, we observed partial dissolution of Sn in the electrolyte, as indicated by ICP-OES analysis of the electrolyte that was collected after 50 min electrolysis (Supplementary Table 4).

For Cu-SnO₂ ALD catalyst, we performed multiple analysis of several samples to ensure the decrease of Sn content (Table R1). According to statistical analysis, the content of Sn indeed decreases after 50 min-CO₂ reduction. However, due to the loading amount of ALD-SnO₂ (~1.35 nm) being quite low, the amount of dissolved Sn is below the detection limit of our ICP-OES. See revised text in Line 141-145.

Table R1 Atomic concentration of Cu and Sn collected on several locations of different Cu-SnO₂ ALD catalysts after pre-reduction and 50 min-CO₂ reduction reaction. The data was detected by SEM-EDX.

Sample condition	Sample point	Cu (at %)	Sn (at %)	Average at % of Sn
After pre-reduction	Sample1-location 1	97.06	2.94	3.02 ± 0.17
	Sample1-location 2	97.06	2.94	
	Sample2-location 1	97.08	2.92	
	Sample2-location 2	96.72	3.28	
After 50 min-reaction	Sample1-location 1	98.31	1.69	1.72 ± 0.68
	Sample1-location 2	98.23	1.77	
	Sample2-location 1	98.23	1.77	
	Sample2-location 2	98.37	1.63	

12. In Table S5, the total FEs of Cu are significantly deviated from 100% from 20 to 100 mA cm⁻² current density, which are not observed on Sn/Cu and SnO₂/Cu. Why did it happen?

Indeed the referee addresses an interesting phenomenon that was also observed in numerous previous studies using a flow reactor: a) Dinh et al. Science 2018, 360, 783; b) Wang et al. Nature Catalysis 2020, 3, 98; c) Hoang et al. JACS 2018, 140, 5791; d) Chen et al. ACS Catalysis 2020, 10, 672. Recently Wang et al. reported that a large fraction of current was used for producing aldehyde in CO reduction at low overpotentials (Nature Catalysis 2020, 3, 98) and volatile liquid products such as aldehydes may escape with the CO flow, leading to an underestimation for liquid products. However, aldehydes are not easily formed during CO₂ reduction.

Another reason is the crossover of ionic products such as formate and acetate from cathode to anode due to the use of anion exchange membrane (AEM). To address this point, we replaced the AEM with a cation exchange membrane (CEM) and carried out CO₂ electroreduction on the Cu catalysts under constant current densities from -20 to -100 mA cm⁻². The additional data is summarized in Table R2. Interestingly, the Faradaic efficiency of HCOO⁻ significantly increased and the total Faradaic efficiency falls into a validated range (84%-93%). Since the selectivity of both Cu-SnO₂ SER and Cu-SnO₂ ALD catalysts dramatically shifts to gaseous CO, the total FE is not much affected due to the usage

of the AEM. New data is included as Supplementary Table 6 and 8. New discussions are from Line 156 to 159.

Table R2. Average faradaic efficiency of the detected products from CO₂ electroreduction on the Cu substrate at different current densities using cation exchange membrane.

Current density (mA cm ⁻²)	Faradaic efficiency (%)								
	H ₂	CO	CH ₄	C ₂ H ₄	HCOO ⁻	CH ₃ COO ⁻	C ₂ H ₅ OH	C ₃ H ₇ OH	Total
-20.0	62.73	9.93	0	1.27	9.63	0.14	0.00	0.00	83.70
-30.0	55.14	12.78	0	4.68	15.64	0.25	0.00	0.00	88.49
-40.0	39.29	15.00	0	11.69	14.73	0.35	2.29	1.81	85.15
-50.0	24.49	22.42	0.01	17.67	14.05	0.32	4.01	3.42	86.40
-60.0	18.92	20.03	0.01	21.42	13.63	0.42	6.30	4.94	85.69
-70.0	17.02	17.90	0.04	23.83	11.50	0.44	8.04	5.27	84.04
-100.0	15.02	13.23	0.08	39.15	9.12	0.42	10.28	5.59	92.89

13. Similar to Pb UPD, the CO adsorption/stripping features are also weird. If the authors look into the referred works by Hori, they can find the CV with or without CO should share similar base lines, and the curves should not be so tilted.

As mentioned in our answer to Q7, the carbon component and porous property of the gas diffusion electrode result in the tilted CV curves with noticeable background current densities in different reaction conditions. However, to ensure that the CO adsorption features represent the real case, we choose to use the data from the samples on a GDE-based flow configuration. We have tried out best to ensure a clear baseline. See revised texts from Line 237 to 241.

14. I do not agree with the discussions in line 289 to 303. The two *CO bands above 2000 cm⁻¹ are both CO_{atop} without any doubt. *CO-bridge has much lower wavenumbers than CO_{atop} (e.g., 1800-1900 cm⁻¹ in neutral pH, J. Am. Chem. Soc. 2020, 142, 6, 2857–2867; ACS Catal. 2018, 8, 8, 7507–7516). The presence of two close CO_{atop} peaks are likely originated from different structures of adsorption sites (J. Phys. Chem. C 2017, 121, 22, 12337–12344; 2019, 9, 1, 474–478).

We thank the reviewer for citing these relevant references. We have corrected the assignment of Raman peak at ~2010 cm⁻¹ to the low frequency CO_{atop} band and a new discussion is now included from Line 291 to 298. A slight red-shift of the CO wavenumber is believed to be attributed to the weaker binding of CO on the SnO₂ doped Cu defect sites (J. Phys. Chem. C., 2016, 120, 15, 8227-8231; Surface Science, 1984, 138, 1, 75-83; Progress in Surface Science, 1985, 19, 4, 275-349). After decoupling the C≡O stretching band into two peaks with Raman shift of ~2010 and 2055 cm⁻¹, a greater intensity ratio of CO_{atop-LW}/CO_{atop-HW} is observed on Cu-SnO₂ ALD catalyst. It has been well-established that atop-bound CO on terraces produces a C≡O stretching band with ~30 cm⁻¹ lower wavenumber than CO adsorbed on undercoordinated defects sites (Surface Science, 1984, 138, 1, 75-83; Progress in Surface Science, 1985, 19, 4, 275-349). Therefore, we believe that the greater intensity ratio of CO_{atop-LW}/CO_{atop-HW} indicates the appearance of Cu sites with a comparatively higher degree of coordination after depositing the SnO₂ layer.

Then, I also have additional concerns regarding the novelty and significance of this work to the CO₂ reduction community. At first, some highly related works with thin SnO₂ layer on Cu and of high CO selectivity have been reported previously, but they are not cited and discussed in this work (e.g., J. Am. Chem. Soc. 2017, 139, 12, 4290–4293; ACS Catal. 2016, 6, 5, 2842–2851). Although they are not tested in flow cell and are not connected to solar cell, the authors should stress the novelty of this work and

what new insights are provided in the material design part.

*We thank the reviewer for drawing our attention to these publications and apologize for missing some relevant literature. These studies have now been cited and discussed within the context of the Introduction section on Pg 2. Compare to prior art, the novelty of our work arises from the following advances in the material design part. (1) In general, the previously-reported Cu-Sn catalysts in H-cell achieve a current density of a few mA cm⁻². Here we boosted the current density by two orders of magnitude. (2) According to previous studies, it is believed that selective conversion of CO₂ to CO only happens by doping the Cu catalyst with trace amounts of Sn. Once the amount of Sn increases, the selectivity dramatically shifts to formate, as seen on our Cu-SnO₂ ALD catalyst. Here we demonstrate for the first time a dramatic difference between CuO-SnO₂ ALD and CuO-SnO₂ SER catalyst with metallic Sn being first sputtered onto CuO and then SnO₂ is formed by the spontaneous oxygen exchange reaction. This is a new approach to prepare a highly selective CO₂ to CO reduction catalyst which is composed of Earth abundant non-noble metal elements. This newly developed catalyst, despite of being Sn-rich in nature, is still capable of catalyzing CO₂ conversion to CO production over a wide thickness range of the Sn layer from 40 to 120 nm. With this Sn-rich Sn/Cu catalyst, we achieved near quantitative conversion of CO₂ to CO at low overpotentials with a Faradaic efficiency of 98%. (3) For the first time, we discovered that SnO₂ formed via oxygen exchange from CuO substrate allows for the binding of CO via the carbon atom to generate C-bound *CO. This enables it to become active for CO generation, as indicated by Sn K-edge X-ray adsorption analysis as well as electrochemical CO adsorption measurements. (4) With this efficient catalyst, an unprecedented solar-to-CO conversion efficiency is achieved. See revised text from Line 368-375.*

Then, the solar-to-CO efficiency is quite high in this work. I am curious whether such amazing efficiency is mainly related to the high efficiency of the solar cell or system integration? It should be mentioned that CO₂-to-CO conversion at low overpotentials, with high current densities, and with over 95% FE has been achieved in many catalysts.

We appreciate the appraisal from the reviewer on the remarkable solar-to-CO conversion efficiency achieved in this work. We attribute the achieved solar-to-fuel efficiency to both the efficient catalyst and the system, including the solar cells.

The solar-to-fuel (STF) conversion efficiency is defined as:

$$\eta_{STF} = \eta_{solar\ cell} \times \eta_{electrolyzer} = \frac{j_{OP} \times V_{OP}}{P_{input}} \times \frac{FE_{CO} \times E_{CO/CO_2}^0}{V_{cell}} = \frac{j_{op} \times FE_{CO} \times E_{CO/CO_2}^0}{P_{input}}$$

Where j_{OP} and V_{OP} are the photocurrent and photovoltage under operation, P_{input} is the illumination power of incident sunlight (100 mW cm⁻²), E_{CO/CO_2}^0 is the thermodynamic potential of the overall reaction for the production of CO (1.34 V for CO), FE_{CO} is the Faradaic efficiency of CO and V_{cell} is the cell voltage for driving the electrolyzer.

If we decouple the effect of solar cell, both FE_{CO} and V_{cell} are the key factors affecting the overall efficiency. At first, both SnO₂ modified Cu catalysts developed in this study enable almost exclusive production of CO with FE_{CO} close to 100%. Secondly, an overall cell voltage of only 2.0 V is required for reaching a current density from -40 to -60 mA cm⁻² to maintain a 95% faradaic efficiency for CO generation in a two-electrode configuration. This is achieved via optimization of the electrolyzer from various aspects, i.e. electrolyte engineering, membrane and configuration of the reactor etc. This leads to an electricity-to-fuel efficiency of over 63%, which is the critical prerequisite for achieving high solar-to-CO energy conversion efficiency. Moreover, the three-phasic interface formed on the gas

diffusion electrode largely improves the mass transport of CO_2 , allowing a dramatic enhancement of partial current density towards CO production as compared to the traditional H-cell.

As for the solar cell, the photovoltage and photocurrent delivered by the solar cell at the maximum power point (MPP) matches well with the requirement of the electrolyzer. In this work, the performance of the selected $\text{InGaP}_2/\text{InGaAs}/\text{Ge}$ triple-junction solar cell (photovoltage of 2.20 V and photocurrent of 14.43 mA cm^{-2} at the MPP) matches very well with the requirement of our electrolyzer, ensuring efficient CO generation in our PV-EC configuration. Relative discussions are included from Line 317-326.

In addition, since the authors are using a GDL flow cell, the current density used for long-term stability test is too small, only 40 mA cm^{-2} , which can even be achieved in H-type cell. Researchers are more interested in seeing the stability data at hundreds of mA cm^{-2} operation in flow cell (e.g., Nature Communications volume 11, Article number: 593 (2020); Nature Energy volume 5, pages1032–1042 (2020); Nature Energy volume 5, pages684–692 (2020)).

As both $\text{Cu-SnO}_2_{\text{SER}}$ and $\text{Cu-SnO}_2_{\text{ALD}}$ catalysts deliver the best Faradaic efficiency towards CO generation at -40 mA cm^{-2} , thus this current density is chosen for stability measurement. A production rate of 34 mA cm^{-2} towards CO generation was shown even after 130 h-electrolysis. **Both the production rate and stability metric achieved in this work outperform those metrics reported on previous Cu-Sn catalysts in H-cells** (Nat. Energy 2017, 2, 17087, J. Mater. Chem. A, 2016,4, 10710-10718).

Following the referee's suggestion, we also tested the stability of two catalysts at -100 mA cm^{-2} , where the selectivity of CO is $< 90\%$. The additional data is shown in the revised Supplementary information (Figure R9). By periodically removing the precipitated salt in the gas-flow chamber and residual liquid on the back side of the working electrode, both $\text{Cu-SnO}_2_{\text{SER}}$ and $\text{Cu-SnO}_2_{\text{ALD}}$ catalysts exhibit a steady generation of CO during $\sim 120 \text{ h}$ -long-term electrolysis at -100 mA cm^{-2} , with the FE of CO remains above 80% after $\sim 120 \text{ h}$. New data is included as Supplementary Figure 31.

Figure R9. Faradaic efficiency of produced CO and the detected half-cell potential at a current density of -100 mA cm^{-2} during 120-hr electrolysis on (a) $\text{Cu-SnO}_2_{\text{SER}}$ and (b) $\text{Cu-SnO}_2_{\text{ALD}}$ catalysts.

Besides, regarding Sn/CuO prepared by sputtering, the surface Sn nanoparticles are of 50-80 nm in size.

Hence, it is expected that there are no interactions between the surface Sn nanoparticles and Cu nanowire substrate as strain and ligand effects usually diminish within 10 nm. And XAS characterizations show no Sn-Cu bonds. I am curious what is the role of Cu in this catalyst? Whether Sn nanoparticles of similar size also selectively produce CO in this reactor? Finally, the Sn/Cu and SnO₂/Cu catalysts have distinct structures and composition, but their CO selectivity and current densities are similar.

We agree with the reviewer that the Sn nanoparticles and Cu nanowires are in the separated phase for the CuO-SnO₂ SER catalyst, as indicated by both XRD results and XAS analysis. We measured the catalytic performance of the pure 60 nm-thick-Sn layer that directly sputtered onto GDE and a favorable generation of formate with Faradaic efficiency >70% was observed at the applied current density from -30 to 100 mA cm⁻² (Figure R10). As demonstrated by XPS and Pb UPD characterization (Supplementary Table 1-2), Cu-SnO₂ SER shows Sn-rich surface after reaction while the selectivity towards CO production still persists. We hypothesize that Sn nanoparticles in Cu-SnO₂ SER catalyst could also be the active site for CO generation via a synergistic effect in the presence of Cu. By further measuring CO adsorption features on these catalysts, we proposed that the presence of Cu strengthens the C-bound intermediate adsorption at Sn sites, altering the selectivity from HCOO⁻ on bare Sn to CO on Cu-SnO₂ SER catalyst. Discussions are included in Line 196-202 and Line 261-269. New data is included as Supplementary Figure 29.

Figure R10. Faradic efficiencies of the products detected on (a) pure 60 nm-thick Sn and (b) 300-ALD-cycle SnO₂ (corresponding to a thickness of ~81 nm) catalysts.

In combination with the technical issues I mentioned above, overall, I feel the relationship between the catalyst structures and performance is unclear in this work.

We thank the reviewer for the constructive comments, the results of which make this work more consistent and clearer. The relationship between the catalyst structure and performance is now clearly presented by providing either additional experiments or a more detailed discussion based on the published literatures.

Reviewer #2 (Remarks to the Author):

Comments on the manuscript entitled “Solar reduction of carbon dioxide on copper-tin electrocatalysts with energy conversion efficiency near 20%” by Jing Gao et. al.

Comments:

The authors reported that sputtered Sn-modified CuO nanowires (Sn/CuO) for production of CO in flow electrolyzer. Sn-rich Sn/CuO catalysts evidenced by X-ray absorption (XAS) and X-ray photoelectron spectroscopy (XPS) achieved $FE_{CO} > 98\%$, $j_{CO} \sim 177 \text{ mA/cm}^2$, and a stability of > 130 hours for CO production. The selectivity of Sn/CuO shifts from CO to formate upon an increase of surface Sn thickness, resulting from increased binding strength of *OCHO rather than *COOH. The authors also showed the solar-driven CO₂ reduction to CO with InGaP₂/InGaAs/Ge triple-junction solar cell system on Sn/Cu catalysts under simulated air mass 1.5G standard illumination, and achieved energy conversion efficiency near 20%. The idea of this work is quite interesting, but such an idea on CO₂RR catalysts employing the electrochemical cells powered by photovoltaics (PV-EC) system has already been reported, as in “ACS Energy Lett., 2020, 5, 470–476”, and “J. Am. Chem. Soc., 2020, 142, 6878–6883.” The difference seems to be only non-noble Cu-Sn materials instead of noble metals in the previous work. This work is evaluated to be lacked originality in publishing to Nature Communications. I think that this paper is better to transfer to other specialized journals and the comments in below will be needed further consideration for publication.

We thank the referee for the summary of our work. As pointed out by the referee, our work substitutes noble metal elements with earth-abundant ones for preparing the electrocatalysts for CO₂ reduction to CO. With this catalyst, we achieved a new benchmark efficiency for solar-driven CO₂ conversion to CO, which is beyond the state-of-the-art. Apart from the that, the novelties of our work are listed below.

*(1) In general, the previously-reported Cu-Sn catalysts in an H-cell achieve a current density of a few mA cm⁻². Here we boosted the current density by two orders of magnitude. (2) According to previous studies, it is believed that selective conversion of CO₂ to CO only happens by doping the Cu catalyst with trace amounts of Sn. Once the amount of Sn increases, the selectivity dramatically shifts to formate, as seen on our SnO₂/CuO catalyst. Here we demonstrate a dramatic difference between CuO-SnO₂ ALD and CuO-SnO₂ SER catalyst with metallic Sn being first sputtered onto CuO and then SnO₂ is formed by spontaneous exchange reaction. This newly developed catalyst, despite of being Sn-rich in nature, is still capable of catalyzing CO₂ conversion to CO production over a wide thickness range of the Sn layer ranging from 40 to 120 nm. With this Sn-rich Sn/Cu catalyst, we achieved near quantitative conversion of CO₂ to CO at low overpotentials with a Faradaic efficiency of 98%. (3) For the first time, we proposed that Sn sites are likely to be modified with a suitable binding with C-bound *CO under the presence of Cu, making it active for CO generation, as indicated by Sn K-edge X-ray adsorption analysis as well as electrochemical CO adsorption measurements. See revised text from Line 368-375.*

1) In Figure 1, it showed that only Sn (200), SnO₂ (200), and Cu (111) plane in TEM images while the various peaks of Sn, Cu, and even CuO are observed in the XRD results. Also, the author needs to provide SAED patterns of Sn-Cu catalysts.

XRD is a technique for characterizing the lattice of bulk composition of the catalyst while the high-resolution (HR) TEM images only represent several particles of the catalysts. Thus, one or two TEM images may only show several lattice patterns. To address this issue, we have performed more HRTEM on as many particles as possible to present a full picture of all the possible lattices. The new data is

shown in the Figure 1h-I, as well as Supplementary Figure 6.

The SAED patterns of two catalysts after pre-reduction are shown in Figure R11. New data is included as Supplementary Figure 15.

Figure R11. High-resolution TEM images and SAED patterns of (a and b) $\text{Cu-SnO}_2_{\text{SER}}$ and (c and d) $\text{Cu-SnO}_2_{\text{ALD}}$. Scale bars: 10 nm for a and c, 2 nm^{-1} for b and d.

2) In Figure 2, total faradaic efficiency (FE_{total}) of Sn/Cu catalysts showed less than 100% in all ranges of current density scale from 20 to 100 mA/cm^2 . Especially, the FE_{total} of Cu showed around 70%, and thereby the author needs to clearly confirm the results of faradaic efficiency of all products.

A similar point has been raised by referee 1 in her/his Q12. Briefly, the crossover of formate from catholyte to anolyte via anion exchange membrane leads to the lower total Faradaic efficiency on the Cu catalyst under cathodic current density lower than 100 mA cm^{-2} (Table R2). However, on the Sn modified Cu catalyst, the production of formate is suppressed. Thus, the total FE is less affected by this unavoidable effect. New data is included as Supplementary Table 5. New discussions are from Line 156-159.

3) In Figure 2, the author changed the Sn-Cu catalysts during the stability tests because the liquid flooding and the resulting salt precipitations degraded the catalytic properties. In Figure 2g and h, the potential results seem to be unstable during the reaction and how often do you change the Sn-Cu catalysts during the stability tests?

We apologize that our description of the experimental details may have not been clear enough. The catalyst for stability test was never changed, only the precipitated salt was removed periodically (Supplementary Figure 30). During the electrolysis, noticeable amounts of bubbles will reside on the surface, leading to small variations in the tested cell potential. This phenomenon seems to be common in flow electrolyzer (Nature, 2020, 577, 509-513, Proc. Natl. Acad. Sci., 2017, 114, 10560-10565). The observed dramatic changes of the potential, i.e. the ones occurred from 10 to 20 h, are mainly due to the unstable reaction condition caused by the blockage of the gas-flow chamber (Supplementary Figure 30). Once a large fluctuation was observed, we disassembled the cathodic gas flow plate and rinsed it using DI water. A clearer description of the process is included from Line 206-210.

4) In this paper, the result of faradaic efficiency (FEs) on Sn-Cu catalysts was only plotted as a function of current density. To compare the effect of Sn on Cu surface more clearly, I think the results of FEs as a function of applied potentials are recommended.

The Faradaic efficiency of different products is now plotted as a function of potential (Figure R12). The Cu nanowires modified with ALD-SnO₂ display a more positive potential at -0.63 V vs. RHE for the optimum CO production with a FE of 98%. The formation of CO remains dominant with FE > 90% cross the moderate overpotential ranging from -0.59 to -0.77 V. While the Sn/Cu electrode shows 70 mV-larger overpotential, at -0.69 V vs. RHE, for the optimum selectivity of 98% towards CO generation. Poor hydrogen evolution and low formate selectivity are observed on both catalysts at potential > -0.75 V. At potential < -0.75 V, the FE towards H₂ and HCOO⁻ starts to increase. In addition, small amounts of C₂₊ products, i.e. C₂H₄ and C₂H₅OH were also generated on the Cu-SnO₂_{SER} and Cu-SnO₂_{ALD} at negative bias of -0.87 and -0.84 V, respectively. The comparatively lower selectivity of C₂₊ products on the Cu-SnO₂_{ALD} electrode suggests a significant suppression of further reduction of produced CO on the surface Cu sites in the presence of SnO₂. New data is included in Supplementary Figure 21. Relative discussions are from Line 171-177.

Figure R12. Faradaic efficiencies of (a) CO and H₂, (b) HCOO⁻ and (c) C₂H₄, C₂H₅OH and C₃H₇OH generated on (a) Cu-SnO₂_{SER} and (b) Cu-SnO₂_{ALD} catalysts under different potentials.

5) In Figure 3, the author claimed that oxygen from the inner CuO layer on Sn/CuO catalysts migrates toward the outer Sn layer during the Cu reduction, resulting in the formation of metallic Cu-supported SnO₂ structure by XAS results. I think the author needs to explain more clearly and precisely in this results.

The exchange reaction between Sn and CuO is spontaneous as the calculated Gibbs free energy of this reaction is negative (Supplementary Note 1):

$$(\Delta G = \Delta H - T\Delta S = -261 \text{ kJ mol}^{-1})$$

In addition, as indicated by HRTEM image of the as-prepared CuO-SnO₂_{SER} sample (Supplementary Figure 4), Cu and O mixed with Sn nanoparticles, visualizing the occurrence of this spontaneous exchange reaction in our sample. On the other hand, based on the XPS analysis, only small amount of metallic Sn remaining on the catalyst surface, further confirming that the reaction happens on the Sn nanoparticles doped CuO nanowires. During the activation of CuO-SnO₂_{SER} catalyst, CuO is fully reduced to the metallic Cu phase as confirmed by XRD (Figure 1b), XPS (Figure 1c) and in-situ Raman results (Figure 4d). The persistence of SnO₂ formed by exchange reaction was observed in the XPS and XAS measurements. Following the referee's suggestion, we have now included a more detailed explanation in Line 65-67 and Line 90-95.

6) In Figure 4, the results of in-situ Raman showed that the Sn/Cu catalysts showed additional peaks of

CO stretch (2010 ~ 2055 cm^{-1}) compared to Cu catalysts during the reaction. However, they showed similar Cu-CO rotation and stretch peaks. Please explain more details on the effect of Sn and how different results depend on the thickness of Sn layers.

*Raman features arise from the frustrated rotation and stretching modes of Cu-CO appear at ~265-284 and ~348 cm^{-1} on both Cu and SnO_2/Cu catalysts. The presence of SnO_2 on Cu surface leads to a blue-shift of Cu-CO rotation peak from ~265 to ~284 cm^{-1} , demonstrating a higher excitation energy of the *CO restricted rotation on Cu active sites after coating with SnO_2 . The upward shift of this rotational frequency is likely attributed to the electrochemical Stark effect (Proc. Natl. Acad. Sci., 2018, 115, E9261-E9270), which is likely caused by the enhanced electrode potential on the SnO_2/Cu over bare Cu (Figure R13).*

Figure R13. Representative linear sweep voltammograms of Cu, $\text{Cu-SnO}_2_{\text{SER}}$ and $\text{Cu-SnO}_2_{\text{ALD}}$ catalysts in 0.5 M KHCO_3 electrolyte, with a scan rate of 25 mV s^{-1} .

In the revised manuscript, we have corrected the assignment of Raman peak at ~2010 cm^{-1} to the low-frequency CO_{atop} band and a new discussion is now included from Line 291-297. Briefly, a slight red-shift of the CO vibrational frequency is believed to be attributed to the weaker binding of CO on the SnO_2 doped Cu defect sites (J. Phys. Chem. C., 2016, 120, 15, 8227-8231; Surface Science, 1984, 138, 1, 75-83; Progress in Surface Science, 1985, 19, 4, 275-349). After decoupling the $\text{C}\equiv\text{O}$ stretching band into two peaks with Raman shift of ~2010 and 2055 cm^{-1} , a greater intensity ratio of $\text{CO}_{\text{atop-LF}}/\text{CO}_{\text{atop-HF}}$ is observed on the SnO_2/Cu catalyst. It has been well-established that atop-bound CO on terraces produces a $\text{C}\equiv\text{O}$ stretching band ~30 cm^{-1} lower than CO adsorbed on undercoordinated defects sites (Surface Science, 1984, 138, 1, 75-83; Progress in Surface Science, 1985, 19, 4, 275-349). Therefore, we believe that the greater intensity ratio of $\text{CO}_{\text{atop-LF}}/\text{CO}_{\text{atop-HF}}$ indicates the appearance of Cu sites with a comparatively higher degree of coordination after depositing the SnO_2 layer. See revised text from Line 291-298.

*Following the referee's suggestion, we also carried out in situ Raman spectroscopic test of 20-cycle-ALD SnO_2 (~5.4 nm) coated CuO catalyst (Figure R14). This catalyst shows highly selective conversion of CO_2 to HCOO^- (Supplementary Figure 28). With the applied current density shifts from -10 to -100 mA cm^{-2} , the features associated with frustrated rotation of *CO, Cu-CO stretching and $\text{C}\equiv\text{O}$ stretching appear at the same position as compared to 1.35 nm-thick SnO_2 modified Cu catalyst, with the peak of $\text{C}\equiv\text{O}$ stretching becoming less intense. Interestingly, an additional broad band of C-H stretching mode appears at ~2900 cm^{-1} at the applied current densities from -20 to -100 mA cm^{-2} . This Raman feature is previously identified as the C-H stretching vibration of the adsorbed formate on Cu*

active sites (*ACS Appl. Mater. Interfaces* 2018, 10, 34, 28572-28581; *ACS Catal.* 2020, 10, 6, 3871-3880) and the appearance of this Raman signal on the thick SnO₂ layer doped Cu is in excellent agreement with its high selectivity towards HCOO⁻ formation (Supplementary Figure 28). These results indicate that the SnO₂ overlayer with different thickness could tune the adsorption energy of different intermediates on Cu active sites, i.e. *CO and HCOO*. New data is included as Supplementary Figure 42 and relative discussions are now included from Line 299-311.

Figure R14. In-situ Raman spectra of 20-cycle-ALD-Cu-SnO₂ ALD catalysts during CO₂ reduction under different current densities.

Reviewer #3 (Remarks to the Author):

The article by Gao and Li et al., explores Sn incorporation on the copper oxide support to evaluate its ability for CO₂ to CO conversion. The authors have further extended this work to plug in a solar cell to the electrolyzer to demonstrate direct solar-to-fuel conversion ability of the catalyst. The results are interesting and I recommend it to be published after minor revision and answering the following questions and comments:

We thank the referee for the positive comment. All the questions and comments have now been addressed by additional experiments and discussions as detailed below.

1. The introduction has appropriately covered the background however, there is still a need to briefly present, which material is the current state-of-art in the conditions explored by the authors. This can span materials used for PV-EC for CO₂ to CO conversion and also the ones, which show the potential from single cell studies for completeness.

Following the referee's constructive suggestion, we have now added a discussion on some highly related studies in the Introduction section of the revised manuscript from Line 40-50.

2. The active catalyst explored in the present article is also the catalyst which was reported as active and selective in their previous work (Reference 16). Why is the overall outcome in the respective PV-EC setup different? Is it only due to pairing with a different anode and electrolyte? Comment on it and also include it explicitly in the text with explanations.

We appreciate the appraisal from the reviewer on the remarkable solar-to-CO conversion efficiency achieved in this work. We attribute the high solar-to-fuel efficiency to both the efficient catalyst and the system, including the solar cells.

The solar-to-fuel (STF) conversion efficiency is defined as:

$$\eta_{STF} = \eta_{solar\ cell} \times \eta_{electrolyzer} = \frac{j_{OP} \times V_{OP}}{P_{input}} \times \frac{FE_{CO} \times E_{CO/CO_2}^0}{V_{cell}} = \frac{j_{op} \times FE_{CO} \times E_{CO/CO_2}^0}{P_{input}}$$

Where j_{OP} and V_{OP} are the photocurrent and photovoltage under operation, P_{input} is the illumination power of incident light (here is 100 mA cm^{-2}), E_{CO/CO_2}^0 is the thermodynamic potential of the overall reaction for the production of CO (1.34 V for CO), FE_{CO} is the Faradaic efficiency of CO and V_{cell} is cell voltage. Under operation, $V_{cell} = V_{op}$.

If we decouple the effect of solar cell, both FE_{CO} and V_{cell} are the key factors affecting the overall efficiency. At first, both SnO₂-modified Cu catalysts developed in this study enable almost exclusive production of CO with FE_{CO} close to 100%. Secondly, an overall cell voltage of only 2.0 V is required for reaching a current density from -40 to -60 mA cm⁻² in order to maintain a 95% faradaic efficiency for CO generation in a two-electrode configuration. This is achieved via optimization of the electrolyzer from various aspects, i.e. electrolyte engineering, membrane and configuration of reactor etc. This leads to an electricity-to-fuel efficiency of over 63%. This efficiency the critical prerequisite for achieving high solar-to-CO energy conversion efficiency. Moreover, the three-phasic interface formed on the gas diffusion electrode largely improves the mass transport of CO₂, allowing a dramatic enhancement of partial current density towards CO production as compared to the traditional H-cell.

In this work, the performance of the selected InGaP₂/InGaAs/Ge triple-junction solar cell (photovoltage of 2.20 V and photocurrent of 14.43 mA cm⁻² at the MPP) matches very well with the requirement of our electrolyzer, ensuring efficient CO generation in our PV-EC configuration. Relative discussions are included from Line 317-326.

3. How do the authors exclude the presence of hydroxylated Sn and Cu species, especially after pre-reduction of CO₂RR? This aspect needs to be addressed in the text or else the possibility should be mentioned along with its plausible influences.

For the as-prepared samples, we suppose that all the hydroxylated Cu(OH)₂ was transformed into CuO component during 1 h-annealing process, as indicated by the absence of peaks ascribed to Cu(OH)₂ in the XRD pattern (Figure 1b). Also, the coating of either metallic Sn or SnO₂ using sputtering and atomic layer deposition will not introduce hydroxide (OH), ruling out the presence of hydroxylated Sn species on the as-prepared samples.

CO₂ reduction will induce the accumulation of OH⁻ ions near the catalyst surface, leading to the possible presence of hydroxylated species. However, we did not observe any hydroxylated component in the XRD patterns of the three catalysts after pre-reduction (Figure 1b). To probe the formation of possible hydroxylated species during reaction, we performed in situ Raman spectroscopy. It was found that a very small peak appears at the Raman shift of 510-515 cm⁻¹ on the surface of Cu substrate as well as 20-cycle-ALD SnO₂ coated Cu catalyst (Figure R15). This Raman peak might be assigned to hydroxyl intermediates adsorbed on electrode surface (ChemElectroChem 2021, 8, 1478). Furthermore, it was reported that the hydroxyl species are beneficial to CO₂ adsorption in the form of H₂CO₃ via hydrogen bonds, and then H₂CO₃ could transform to HCOO* species (J. Am. Chem. Soc. 2019, 141, 7, 2911-2915, J. Catal. 2016, 343, 257-265). This is in an excellent agreement with the favorable formation of HCOO⁻ on these two catalysts. Please see the revised text from Line 305-308.

Figure R15. In-situ Raman spectra of (a) Cu and (b) 20-cycle-ALD-Cu-SnO₂_{ALD} catalysts during CO₂ reduction under different current densities.

4. How do the authors envision the mechanism of spontaneous displacement between Sn and CuO mentioned? Is it self-limiting or extends to the near surface region or any other scenario?

Theoretically, the replacement reaction between Sn and CuO is spontaneous since the overall reaction has a negative ΔG value (Supplementary Note 1):

The more strongly reducing metal (Figure R16), Sn in our case, is capable of displacing the less reactive metal from its oxide lattice. As indicated by the high-resolution TEM image of as-prepared CuO-SnO₂_{SER} sample, Cu and O distribute where Sn nanoparticles are located, visualizing the occurrence of this spontaneous exchange reaction in the CuO-SnO₂_{SER} sample (Supplementary Figure 4).

Based on our XPS and XAS analysis, most of the Sn species remain as oxides. This indicates that sputtered Sn metal reacts with CuO and becomes SnO₂. Please see revised text from Line 65-67 and Line 90-95.

Figure R16 Reactivity series of typical metals (Twidle, J. & Crowley, M. The reactivity series. Secondary Science 11 to 16: A Practical Guide, 2010, 49)

5. What is the peak in XRD of as-prepared sample around same 2θ as for Cu(111) after pre-reduction shown in Figure 1b?

The XRD peak at $2\theta = 43.2^\circ$ arises from the polytetrafluoroethylene (PTFE) component of the gas diffusion electrode, as indicated by the reference added to the revised Supplementary Figure 1. Additionally, the XRD patterns of the catalysts deposited on etched FTO glass are now shown in the revised Figure 1b to eliminate the effect of the GDE substrate on the diffractive peaks.

6. Does the sputter deposition of Sn allow for intermixing of Sn and Cu rather than just coating the top surface? Can this aspect be the reason for catalyst performance versus previous reports mentioned in line 196-198?

The intermixing between Sn and Cu might be possible. However, the XRD of $\text{Cu-SnO}_2_{\text{SER}}$ catalysts shows no peak that could be ascribed to Cu-Sn alloy (Figure 1b). Our XAS analysis of $\text{Cu-SnO}_2_{\text{SER}}$ catalyst after pre-reduction also reveals that no Cu-Sn bond is observed on the surface of this sample (Figure 3f). Based on our analysis, we conclude that there is no observable interphase mixing between Cu and Sn in our $\text{Cu-SnO}_2_{\text{SER}}$ system. See revised text in Line 116.

7. Are the XAS studies performed in situ or ex situ?

In this work the XAS measurements of all the catalysts were performed ex situ, but all the samples were immediately packed under vacuum after the sputtering or pre-reduction process. Since both Cu and Sn are easily oxidized during air exposure due to their oxophilicity, the vacuum packaging helps to avoid the impact of atmospheric oxidation at the catalyst surface and ensures that its composition and structure represent their real feature formed under operational conditions. See revised text from Line 474-484.

8. The fitting of post-edge XAS data points towards the presence of Sn-Sn bond for one case and Sn-Cu for the other. Since different components are considered for the fitting of the two catalysts, how does the fit turn out considering similar components?

We indeed applied the same structure (Sn-doped Cu_6SnO_8 : <https://materialsproject.org/materials/mp-1147658/>) for the EXAFS fittings of $\text{Cu-SnO}_2_{\text{ALD}}$ and $\text{Cu-SnO}_2_{\text{SER}}$. Using the FEFF6 program (included in Demeter (v.0.9.26): <https://bruceravel.github.io/demeter/>), we found that the model structure underwent different relaxations to form metallic Cu-Sn bond in the $\text{Cu-SnO}_2_{\text{ALD}}$ catalyst and metallic Sn-Sn bond in the $\text{Cu-SnO}_2_{\text{SER}}$ catalyst, respectively. Schematics of materials relaxations are shown below:

Figure R17. Schematics of model structure relaxations during EXAFS fittings.

Relative descriptions of this process are included from Line 472-474 in the revised manuscript and Supplementary Figure S32.

9. Given XAS shows the presence of different sites on the two catalysts explored, why is the selectivity for both similar towards CO₂RR (line: 231-234)? Typically, the presence of Sn-Sn should result in more formate.

It is generally believed that selective conversion of CO₂ to CO could be achieved by doping the Cu catalyst with trace amounts of Sn (J. Am. Chem. Soc. 2017, 139, 12, 4290-4293; Nat. Energy 2017, 2, 17087; Adv. Energy Mater. 2022, 12, 2103328). Once the Sn content was slightly increased, the selectivity of the catalyst dramatically shifts to formate. This phenomenon is also validated by the SnO₂ modified CuO nanowires developed in this work (Supplementary Figure 27). By in situ Raman spectroscopic measurement, we revealed that the SnO₂ overlayer is capable of tuning the adsorption of key intermediates on the Cu active sites, which agrees with previous reports (Adv. Energy Mater. 2022, 12, 2103328, ACS Catal. 2019, 9, 10, 9411-9417, Chem. Commun. 2018, 54, 13965-13968).

*Despite of having a Sn-rich surface composition, Cu-SnO₂ SER, is still capable of catalyzing CO₂ conversion to CO, but Sn nanoparticles deliver a favorable HCOO⁻ formation (Figure R18). By assessing in situ CO adsorption behavior on the catalyst surface, we for the first time proposed that Sn sites are likely to be modified with a suitable binding with C-bound *CO under the presence of Cu, making it active for CO generation.*

Figure R18. Faradic efficiencies of the products detected on (a) pure 60 nm-thick Sn and (b) 300-ALD-cycle SnO₂ (corresponding to a thickness of ~81 nm) catalysts.

10. CVs in Figure 4b and S32c (add label a,b,c in the figure) for Sn/Cu shows a capacitive behavior perhaps due to rough vs. near-planar catalyst. How to decouple the effect of CO₂ being trapped aiding intermixed Sn-Cu to allow further reduction versus more uniform SnO₂ from ALD? Besides, the authors point out about the CO adsorption on Sn being intense in the presence of Cu (line: 264-266) but the desorption peak on Sn or SnO₂ is quite intense compared to Sn/Cu, which cannot be the case unless they also adsorb CO. Comment on it.

From our measurements of double-layer capacitance of both catalysts, the Cu-SnO₂ ALD electrode was assessed to be rougher than Cu-SnO₂ SER (Supplementary Figure 13 and Table 2). Even though some CO₂ molecules might be trapped by the Cu-SnO₂ ALD catalyst due to its larger capacitance, the porous gas diffusion electrode used in this work allows efficient transport of CO₂ and the trapped CO₂ makes insignificant contribution to the electrocatalytic performance of different catalysts.

The desorption peak of CO on the Cu-SnO₂ SER catalyst is likely to be affected by the capacitive behavior of the Cu substrate, making it look less intense as compared to the one recorded on the pure

Sn and SnO₂ surface. However, the desorption charge on the Sn/Cu is assessed as 0.18 mC cm⁻², which is 3× higher than the one on the Sn catalyst (0.065 mC cm⁻²). See revised text in the caption of Supplementary Figure 39.

11. The catalyst characterization has been carried out in 0.5 M KHCO₃ throughout the entire work but the final PV-EC application is studied in 2M KOH. How can the understanding of the catalyst in a near neutral media be directly used to explain the outcome of functioning in a highly basic electrolyte? How is the catalyst performance in the PV-EC configuration using 0.5 M KHCO₃ instead?

An alkaline KOH electrolyte instead of near-neutral KHCO₃ electrolyte was used for photosynthesis system for the purpose of decreasing the overpotential for cathodic CO₂ reduction (Electrochimica Acta 2015, 166, 271-276). The higher concentration of 2 M will also decrease the ohmic loss between the two electrodes. Thus, 2 M KOH is supposed to decrease the required cell voltage in the unbiased system. As suggested by the referee, we also carried out the PV-EC CO₂ reduction using 0.5 and 2 M KHCO₃ electrolytes, with additional results shown in Figure R19.

Figure R19. (a and d) Linear sweep voltammograms of the solar cell (orange) and electrolysis cell (blue) in 0.5 M and 2 M KHCO₃ electrolyte, here the current density of electrolysis is normalized against working area of solar cell. The performance of photovoltaic is measured under simulated standard AM 1.5G with intensity of 100 mW cm⁻². (b and e) Solar current density and voltage of the unassisted PV-EC system under standard AM 1.5G illumination when the 0.5 M and 2 M KHCO₃ are used as the electrolyte. (c and f) Faradaic efficiency of produced CO and STF conversion efficiency delivered by the unassisted PV-EC system under standard AM 1.5G illumination.

In the two cases, the unassisted systems delivered a predicted operating point far from the maximum power output metric of the solar cells. By further characterizing the performance without applying any external bias under AM 1.5G light illumination, average current densities of only -3.5 and -6.5 mA cm⁻² were reached in 0.5 M and 2 M KHCO₃ electrolyte, leading to a Faradaic efficiency of 59.2% and 90.4% for CO production, respectively. As a result, the solar-to-CO conversion efficiency was calculated as only 2.5% and 7.5% using 0.5 M and 2 M KHCO₃ electrolytes, respectively. These

additional data is included as Supplementary Figure 45 and the corresponding discussion are also included in the revised manuscript in Line 341-344.

12. XPS survey scan of the two catalysts coated on GDE should be provided to confirm absence of any other contributions.

The XPS survey scan of the as-prepared samples as well as the ones after pre-reduction are shown in Figure R20. As compared to the spectra of Cu substrate, no detectable peaks from elements other than Sn was observed with the Sn and SnO₂ modified Cu catalysts, ruling out the presence of any contaminants on the catalyst surface. New data is included in Supplementary Figure 2 and 11.

Figure R20. XPS survey scan of (a) as-prepared samples and (b) pre-reduced catalysts. The pre-reduction was performed at 30 mA cm⁻² for ~100 s.

13. What are the additional peaks in XRD of Fig. S1(a & b) for GDE? Peaks for CuO and Sn are barely visible, zoomed insets or separate figures are recommended along with goodness of fit for the claims. An alternative can be catalyst deposition on another planar substrate (glassy carbon for instance) to confirm the claims by eliminating GDE influence. Are XRD peaks for Sn and SnO₂ are expected at the same 2θ? Fig. S1c, a Shirley background is used for SnO₂/CuO and linear for the other two cases, comment on it. Additionally, binding energy scales for XPS is typically represented from higher to lower values on the x-axis.

The commercial GDL used in this work is based on carbon paper covered by a hydrophobic polytetrafluoroethylene (PTFE) layer (<https://www.fuelcellstore.com/sigracet-38-bc>). The peaks shown in the diffractogram of GDL are now assigned to carbon (PDF #00-026-1076), (C₂F₄)_n (PDF #00-061-1414) and CF₄ (PDF #00-055-0072) species from PTFE layer, as indicated by the reference added in the revised Supplementary Figure 1. Moreover, we also prepared the catalysts on glass substrate. The XRD patterns clearly show Sn(200), Sn(101), Sn(211), CuO(002), Cu(111) and Cu(200) for the as-prepared samples and Sn(200), Sn(101), Cu(111) and Cu(200) for the pre-reduced catalysts. This result demonstrates that no metal contaminant is present in our catalyst. Following the reviewer's comment, we have also provided the XPS spectra of a bare GDL in the Supplementary Figure 2 in the revised Supplementary Information. New discussions are now included from Line 72-80.

In the revised manuscript, all the XPS spectra have been updated using quasi in situ measurement method. The spectra have been plotted from higher to lower binding energy and all the high-resolution XPS plots are fitted with background type of linear or universal cross-section Tougaard using Casa XPS software. See updated Figure 1 and Supplementary Figures 2 and 11.

14. Fig. S7(f): Zoomed TEM showing SnO₂ layer around 1.3 nm is recommended to support the

outcome from Fig. S5.

A high-resolution HAADF image as well as EDX mapping of Cu, Sn, O are now shown in Supplementary Figure 5. However, it is still challenging to determine the exact thickness of SnO₂ layer via HAADF-EDX. Thus the thickness of SnO₂ is determined via a linear fitting between thickness and deposition cycles via the equation:

$$d_{\text{thickness}} = N * d_0 \text{ (} N \text{ represents the number of cycles, } d_0 \text{ represents the thickness per cycle)}$$

The thickness of SnO₂ layer with different deposition cycles (from 200 to 400) was measured by the cross section of planar-CuO-supported SnO₂ using a focus ion beam (Supplementary Figure 8). According to the calibration curve of the thickness as a function of deposition cycles (Supplementary Figure 9), the thickness of 5-ALD-cycle SnO₂ film was estimated to be 1.35 nm.

15. Fig. S9: Same x-y scaling is recommended for better comparison, provide zoomed insets or separate figures wherever necessary. What is the reason for slanted background in the CVs?

The cyclic voltammograms of Pb UPD are re-measured as suggested and all the updated curves are plotted using the same x-y scale for clear comparison between different electrodes (Figure R5).

In order to unravel the real surface condition of each catalyst, all the electrochemical measurements including double-layer capacitance, Pb UPD and CO adsorption were carried out in the same flow configuration used for running CO₂ electroreduction. We observed that all the CV curves are tilted even after removing possible existing oxygen by flowing He into the sealed reactor as well as using He-saturated electrolyte. Note that the gas diffusion electrode used in this work is very porous and residual oxygen may exist inside the pores. This leads to the tilted CV curves with noticeable background current densities, as indicated by the CV measurement of Cu films deposited onto different substrates. Since we are using the same GDE for preparing different catalysts, we assume that the interference from the substrate remains similar when we compare the electrochemical properties between different catalysts.

16. Fig. S11: Electrolyte used for the measurement is not mentioned. Also, CV in a wider potential ranged should be shown to justify no/minimal interference from faradaic currents. What is the reason for noisy traces in Fig. S11(c)? Also, what are the redox features observable in Fig. S11(b) for higher scan rates? How was the roughness factor computed? What value did the authors use for a flat surface and why?

For assessing the ESCA of each catalyst, the electrolyte of 0.5 M KHCO₃ was flowed into the chamber at a rate of 0.25 cm³ min⁻¹. More details are now added in the Experimental section.

The observed noisy traces may arise from electromagnetic noise. To rule out the interference, we re-scanned the cyclic voltammograms on three electrodes (Figure R8) and the curves in a wide potential range from 0.35 to 0.85 V vs. RHE for Cu and 0.25 to 0.95 V vs. RHE for Cu-SnO₂_{SER} and Cu-SnO₂_{ALD} are also provide in the Figure R8.

The voltage range we chose for double layer capacitance measurement of the three samples falls in a non-faradaic regime. The roughness factor can be calculated by referring the obtained capacity to the reference value of capacity per unit area of a strictly planar electrode surface. In general, the reference value of capacity per unit area (C_{ref}) reported in literatures is typically between 20 to 40 $\mu\text{F cm}^{-2}$ in alkaline solution (Int. J. Electrochem. Sci., 11 (2016) 4442-4469; Pure & Appl. Chem., 1991, 63, 5, 711-734; J. Am. Chem. Soc. 2013, 135, 45, 16977-16987; J. Am. Chem. Soc. 2015, 137, 13, 4347-4357). Here we use an average value of 30 $\mu\text{F cm}^{-2}$ for calculating the roughness factor of Cu and Cu-Sn electrodes. New data is included as Supplementary Figure 13.

17. Amount of Sn is shown to decrease after pre-reduction (Fig. S12 vs. S13), what is the expected cause? Does it go into the electrolyte? Could that be the reason that eventually more Cu is exposed giving the observed activity?

Referee 1 asked the same question in Q11, we kindly ask the referee to refer to our answer to Q11 from referee 1.

According to the SEM image of Cu-SnO₂ SER after 50 min-reduction (Figure R21), SnO₂ particles remain to be densely packed on the surface of Cu nanowires, with insignificant amount of Cu being exposed to the electrolyte. Thus, we propose that Sn instead of Cu is likely to be the active site for CO generation. New data is included as Supplementary Figure 17.

Figure R21. Characterization of Cu, Cu-SnO₂ SER and Cu-SnO₂ ALD catalysts after CO₂ electroreduction under -50 mA cm⁻² for 50 min in 0.5 M KHCO₃ electrolyte. Scanning electron micrographs of (a) Cu, (b) Cu-SnO₂ SER and (c) Cu-SnO₂ ALD after 50 min-CO₂ reduction.

18. Fig. S37: What is the peak around 3.3 eV? Does Sn deposit over the anode?

The peak at 3.3 eV is ascribed to potassium (Supplementary Figure 45), resulting from the residual electrolyte on the electrode. According to our SEM-EDX characterizations, Sn is not present on the surface of IrO_x electrode after the reaction.

19. Fig. S38: Significant Sn is lost during the 3h electrolysis (compared to Fig. S6 & S12). This should be explicitly mentioned in the text with the possible causes.

Partial dissolution of both Cu and Sn was confirmed by additional ICP-OES analysis of the collected electrolyte after ~3 h solar-driven CO₂ reduction. See the revised text from Line 361 to 365.

20. SI Table S1: How did the authors ensure removal of K from the catalyst after pre-reduction and before XPS analysis?

We have carefully removed any residual ions on the electrode right after pre-reduction. The liquid chambers of the electrochemical flow cell were flushed with DI water to remove the residual potassium and bicarbonate ions to prevent the possible reaction between the ions with the catalyst during the disassembling of the cell. The sample was then removed from the cell. Both the front and back of the gas diffusion layer were washed with DI water thoroughly, followed by drying with an air gun. The washing process was repeated three times to remove residual ions on the surface. With these precautions during the preparation of samples, we did not observe features showing significant residual potassium. Details are now described in the 'Experimental section' in the revised manuscript from Line 472-484.

21. SI Table S2: Mention the background assumed for charge integration. Comment on the interaction of Pb and Sn during Pb-UPD.

As seen in the updated cyclic voltammograms for Pb UPD (Supplementary Figure 12), the control curves measured in bare HClO_4 show a small anodic current close to zero mA cm^{-2} on all three catalysts, thus we used zero as the background for charge integration.

The referee is correct in pointing out the possible interaction between Pb and Sn nanoparticles during Pb UPD measurement. At first, we did not find any related study that has reported underpotential deposition of Pb on Sn substrate. We then performed Pb UPD on the sputtered Sn electrode in the same system used for Sn/Cu catalysts. As shown in Figure R22, no feature due to underpotential deposition or stripping of Pb was observed on Sn electrode. From this we infer that there is no interaction between Sn and Pb during the underpotential deposition of Pb on our catalysts, ruling out the possible interference from Sn on the Pb monolayer formation and stripping on the Cu active sites. The discussions are now included in the revised Supplementary Information Figure 12.

Figure R22 Representative cyclic voltammograms for possible underpotential deposition of Pb on bare Sn electrode.

22. SI Table S4: SnO_2/Cu catalyst shows no Sn detected in ICP but EDS shows a decrease in Sn content from 2.6% to 1.5% post electrolysis. Comment on it.

Referee 1 raised the same concern in Q11, we kindly ask the referee to refer to our answer to Q11 from referee 1.

23. Revise the paper for minor typos like: (a) Line 79: X-ray diffraction (vs. X-ray diffractive); (b) Line:243-244 not clear, rephrasing will be better; (c) denotation of the catalyst is many times reversed

which is confusing, make it consistent. Examples: Fig 4a (Cu/OD Sn); Fig 4b (Cu/Sn); Fig 5b (Cu/OD Sn); Fig. S7 (CuO/Sn); Fig. S16 (Cu/SnO₂); Fig. S18a; (d) Line 283: When instead of with the reduction current...; (e) Line 300: coating (vs. coting); (f) Line 314: PV-EC (vs. PV-FC); (g) Line 555: quantify (vs. qualify); (h) SI Line 54-55: E_{fuel} (vs. E_{fule}); (h) SI Line 93-94: Incorrect label description for Fig. S4.; (i) Represent either by EDS or EDX uniformly throughout the text; (j) SI Line 186-188: Meaning not clear, rephrase; (k) Fig. S29: He-trace barely visible; (l) Several instances in the text show different representation of the triple-junction solar cell (for example: Fig. S35 image & caption; SI Table S13) is that intentional, were different configurations used?; (m) SI Line 326: Scale (vs. Scar); (n) Fig. 5b: MPP is not explained anywhere in the text; (o) Line 378: strategy to produce (vs. strategy produce).

We appreciate the referee's comments on those typos. Following the referee's comments, we have addressed all these issues in the revised manuscript. We have also checked through the manuscript and corrected details where necessary.

Peer review comments,

second round review

Reviewer #1 (Remarks to the Author):

Regarding my previous comment 7, the Pb UPD curves are still weird. Ideally, the Pd monolayer deposition and Pd monolayer stripping segments should give similar results on the surface area of Cu, but in the CVs provided by authors, the difference would be very large. Especially the Cu-SnO₂ SER, the ECSA calculated by the Pd monolayer stripping segment was even negative. The error of this characterization is too large for scientific analysis.

Other issues have been addressed.

Reviewer #3 (Remarks to the Author):

The authors have performed additional experiments and have answered the questions raised during the review process. I recommend its publication but after answering two persistent questions:

(1) A question raised by me (Reviewer 3: Q9) was also raised by Reviewer 1 (Q14: 5th part) i.e., both Sn/Cu and SnO₂/Cu have different structure, composition and bonding but why there selectivity and current density are similar towards CO? This has not been addressed so far and is critical.

(2) The authors have clearly demonstrated the benefit of using KOH vs. KHCO₃ electrolyte for the PV-EC configuration but how do they prove that the Cu-Sn (in Cu-SnO₂ ALD) and Sn-Sn (in Cu-SnO₂ SER) bonding remains unchanged in 2M KOH?

Our point-by-point response to the referees' comments is listed below in blue italic fonts.

Reviewer #1 (Remarks to the Author):

Regarding my previous comment 7, the Pb UPD curves are still weird. Ideally, the Pd monolayer deposition and Pd monolayer stripping segments should give similar results on the surface area of Cu, but in the CVs provided by authors, the difference would be very large. Especially the Cu-SnO₂ SER, the ECSA calculated by the Pd monolayer stripping segment was even negative. The error of this characterization is too large for scientific analysis. Other issues have been addressed.

The referee raised concern regarding the Pb UPD measurement. Here we provided more details and explanations. In general, we respectfully disagree with the conclusion that the error of Pb (instead of Pd) UPD is too large for the scientific analysis. We carefully reprocessed all the representative CV curves of Pb UPD (Figure R1). The charge of both underpotential deposition and stripping has been calculated by integrating the corresponding peaks by Origin software. Table R1 clearly summarized the charges between monolayer deposition and stripping that have similar values instead of large difference on all three catalysts. On the other hand, the Pb UPD was performed to prove that the number of Cu active sites would decrease after coating either SnO₂ SER or SnO₂ ALD and the charges determined here are not used for any quantitative analysis in this work.

Figure R1. Representative cyclic voltammograms showing the stripping and underpotential deposition of Pb on Cu active sites from (a) Cu nanowires, (b) Cu-SnO₂ SER and (c) Cu-SnO₂ ALD catalysts using the electrolyte containing 0.1 M HClO₄ and 0.001 M Pb(OAc)₂, the scan rate is 50 mV s⁻¹ for all the voltammogram tests.

Table R1. Charge from the stripping of Pb on Cu, Cu-SnO₂_{SER} and SnO₂/Sn catalysts estimated from cyclic voltammograms shown in Supplementary Figure 12 and Figure 18, the measurements were carried out in 0.1 M HClO₄ + 0.001 M Pb(OAc)₂ aqueous electrolyte at a scan rate of 50 mV s⁻¹.

Catalysts condition	Catalysts	Charge consumed for Pb stripping on Cu sites (mC cm ⁻²)	Charge consumed for Pb underpotential deposition on Cu
After pre-reduction	Cu	3.81 ± 0.46 ^a	3.54 ± 1.4
	Cu-SnO ₂ _{SER}	0.54 ± 0.04	0.51 ± 0.06
	Cu-SnO ₂ _{ALD}	1.76 ± 0.03	1.925 ± 0.32

^aThe charge consumed for Pb stripping correspond to the average value of two to three independent measurements and the error bars are the standard deviations of these measurements.

Finally, additional cyclic voltammetry has been performed on CuO catalyst that had been deposited on three different substrates: 1) planar FTO, 2) GDE, 3) GDE with back side of carbon paper been covered (Figure R2). We concluded that the carbon component and porous property of the GDE result in the tilted CV curves with noticeable background current densities. However, to ensure that all the features represent the real case, we choose to use the data from the samples on a GDE-based flow configuration. Nevertheless, the peak features of stripping and deposition are clearly recorded in the voltammograms (Figure R1).

Figure R2 Cyclic voltammograms of CuO films prepared onto different substrates: (a) etched planar FTO and GDE with back side of carbon paper was exposed to the electrolyte or covered by epoxy.

In summary, we believe that our Pb UPD measurements are reliable for the scientific analysis performed in this study. Please see text from Line 124-126. Additional data is shown as Supplementary Figure S12.

Reviewer #3 (Remarks to the Author):

The authors have performed additional experiments and have answered the questions raised during the review process. I recommend its publication but after answering two persistent questions:

We thank the referee for the positive comment. Two residual questions have now been addressed by additional discussions and experiments as detailed below.

(1) A question raised by me (Reviewer 3: Q9) was also raised by Reviewer 1 (Q14: 5th part) i.e., both Sn/Cu and SnO₂/Cu have different structure, composition and bonding but why there selectivity and current density are similar towards CO? This has not been addressed so far and is critical.

*For Cu-SnO₂ ALD catalyst, in situ Raman spectroscopy have revealed that SnO₂ overlayer with different thickness is capable of tuning the adsorption energy of key intermediates, *CO and *OOCH, on Cu sites (Figure 4c-d and Supplementary Figure 42). Due to the large Sn-to-Cu ratio of 10:90 on the surface of Cu-SnO₂ ALD by XPS analysis (Supplementary Table 1), we conclude that the Cu would be the predominant active sites for CO production after being modified by SnO₂ ALD.*

*For Cu-SnO₂ SER catalyst, combined with the Sn-rich surface with Sn-to-Cu ratio of 58:42 but still remarkable CO production rate (Supplementary Table 1 and Figure 2c-d), we believe that surface SnO₂ SER is also selectively catalyzing CO production in this case. The CO adsorption/stripping features observed on the surface SnO₂ SER sites further provide evidence that the presence of Cu shifts the adsorption preference of Sn sites to C-bound *CO (Figure 4b).*

In short, SnO₂ ALD modifier assists the CO production on Cu sites while SnO₂ SER has been modified by Cu to be selective towards CO generation. Though the catalytic active sites are different on the two catalysts, both SnO₂ ALD-modified Cu sites and Cu-modified SnO₂ SER sites show excellent selectivity towards CO production. Please see text from Line 268-270.

(2) The authors have clearly demonstrated the benefit of using KOH vs. KHCO₃ electrolyte for the PV-EC configuration but how do they prove that the Cu-Sn (in Cu-SnO₂ ALD) and Sn-Sn (in Cu-SnO₂ SER) bonding remains unchanged in 2M KOH?

In our manuscript, KOH is used for the demonstration of the PV-EC system. Though the question raised here is beyond the scope of our study, we conducted additional experiments to determine the bonding features in alkaline electrolyte.

In solar driven CO₂ reduction system, Cu-SnO₂ ALD was selected as the catalyst due to its lower overpotential (Supplementary Figure 19). The CO adsorption/stripping features have been characterized on both Cu and Cu-SnO₂ ALD surface by flowing aqueous 2 M KOH electrolyte (Figure R3). The peak at -0.23 V vs. RHE is attributed to CO adsorption on the Cu active sites and two closely adjacent peaks at -0.18~-0.15 V are assigned to CO stripping. Similar but faint signals are also observed on Cu-SnO₂ ALD catalyst. These observations are similar with the one measured in 0.5 M KHCO₃. Combined with the high selectivity of Cu-SnO₂ ALD towards CO production (Figure 5d), we believe that the bonding configuration remains unchanged in alkaline electrolyte. Additionally, the local environment has been revealed to be strongly basic with pH near 12 even when a KHCO₃ solution is used as the electrolyte (J. Am. Chem. Soc. 2020, 142, 36, 15438–15444). This is likely to be one of the reasons for the

unchanged catalyst structure and performance in different electrolytes. New discussions are now included from Line 342 to 344. New data is included as Supplementary Figure S45.

Figure R3 Cyclic voltammograms of Cu and Cu-SnO₂ ALD catalysts measured in 2 M KOH electrolyte with He (dash line) and CO (solid line) been flowed at 50 cm³ min⁻¹, the scan rate is 50 mV s⁻¹.